# Obliviator Reveals the Cost of Nonlinear Guardedness in Concept Erasure

**Ramin Akbari**[*]    **Milad Afshari**[*]    **Vishnu Naresh Boddeti**

Michigan State University
`{akbarigh, afsharim, vishnu}@msu.edu`

## Abstract

Concept erasure aims to remove unwanted attributes, such as social or demographic factors, from learned representations, while preserving their task-relevant utility. While the goal of concept erasure is protection against all adversaries, existing methods remain vulnerable to nonlinear ones. This vulnerability arises from their failure to fully capture the complex, nonlinear statistical dependencies between learned representations and unwanted attributes. Moreover, although the existence of a trade-off between utility and erasure is expected, its progression during the erasure process, i.e., the cost of erasure, remains unstudied. In this work, we introduce *Obliviator*, a post-hoc erasure method designed to fully capture nonlinear statistical dependencies. We formulate erasure from a functional perspective, leading to an optimization problem involving a composition of kernels that lacks a closed-form solution. Instead of solving this problem in a single shot, we adopt an iterative approach that gradually morphs the feature space to achieve a more utility-preserving erasure. Unlike prior methods, *Obliviator* guards unwanted attribute against nonlinear adversaries. Our gradual approach quantifies the cost of nonlinear guardedness and reveals the dynamics between attribute protection and utility-preservation over the course of erasure. The utility-erasure trade-off curves obtained by *Obliviator* outperform the baselines and demonstrate its strong generalizability: its erasure becomes more utility-preserving when applied to the better-disentangled representations learned by more capable models.

## 1   Introduction

Pretrained Language Models (PLMs) have become central in Natural Language Processing (NLP), achieving remarkable performance across various tasks. However, they often encode unwanted concepts, such as demographic or social attributes, leading to biases and potentially unfair predictions [31, 5, 26]. Concept erasure aims to remove such undesirable concepts from the learned representations [27], either supervised, leveraging both task and sensitive labels or unsupervised, using only sensitive labels.

**Current concept erasure methods fall short of fully removing undesired attributes**. The task involves balancing two often competing objectives: (1) removing unwanted concepts, and (2) preserving as much information as possible. Early methods relied on linear projection techniques [5, 15, 26, 27] applied in a post-hoc manner, i.e., modify learned representations without needing direct access to the PLMs, offering simplicity and efficiency. Later methods shifted to addressing concept erasure in a nonlinear manner [28, 30, 1, 6, 2]; however, they remain limited in capturing

---

[*]Equal contribution.

39th Conference on Neural Information Processing Systems (NeurIPS 2025).

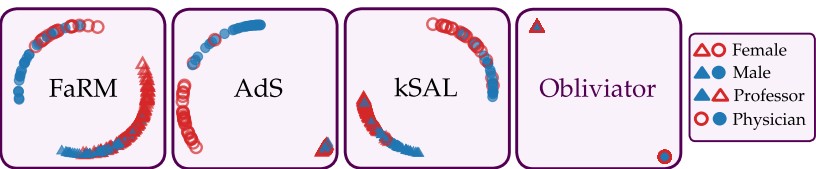

Figure 1: **Erasure of Gender from Representation on BIAS IN BIOS**. Embeddings from a nonlinear adversary trained to extract gender information from the erased representation. Existing nonlinear methods fail to fully protect gender, as gender-specific distributions within each profession remain distinguishable. In contrast, Obliviator effectively guards gender by overlapping representations across gender, while preserving separability by profession.

nonlinear dependencies and are vulnerable to nonlinear adversaries. Even with costly fine-tuning of PLMs, as we will show in this work (§5), erasure is not complete. Moreover, the dynamics of the competition between objectives, i.e., the progressive loss and gain in each objective throughout the erasure process, and how they shape the resultant utility-erasure trade-off, have not been explored.

**This paper studies the cost of nonlinearly guarded erasure.** We propose Obliviator, which performs erasure by minimizing the statistical dependency between learned PLM representations and unwanted attributes. Our formulated erasure problem solves a multi-objective optimization with competing objectives. Solving this optimization in a single shot is challenging and likely leads to a poor solution. Instead, Obliviator solves this problem using a two-step iterative procedure, enabling us to gradually morph the feature space to obtain a more utility-preserving erasure. We utilize the Reproducing Kernel Hilbert Space (RKHS), which allows us to capture nonlinear statistical dependencies in a closed form, facilitating the optimization. Our stable optimization approach provides a reliable way to study the dynamics of the objectives during the erasure process, known as the utility-erasure trade-off. Through these trade-off curves, across various experiments, we demonstrate Obliviator outperforms baselines while guarding against nonlinear adversaries (see Figure 1). Moreover, its erasure becomes more utility-preserving when applied to the better-disentangled representations learned by more capable PLMs.

**Summary of Contributions**:

1. We introduce Obliviator, a post-hoc erasure method that captures the nonlinear dependency of sensitive attributes to original representation. Obliviator guards unwanted attributes against nonlinear adversaries.(§4)

2. Our stable optimization approach provides a reliable way to study the dynamics of the utility-erasure trade-off. Across all experimental setups, Obliviator consistently achieves higher task performance at every level of erasure, setting a strong benchmark for this trade-off. (§5.1)

3. We illustrate the generalizability of Obliviator through experiments across various language models. We observe that its erasure becomes more utility-preserving when applied to the better-disentangled representations learned by more capable PLMs.(§5.2)

## 2   Related Works

**Removal of sensitive attributes from learned representations** was first explored in the context of fair classification [33]. The task, often termed concept erasure, gained prominence with applications such as removing binary gender labels from GloVe embeddings [22], and has since evolved into a broader research area at the intersection of fairness, interpretability, and adversarial representation learning. Existing approaches can be categorized as *post-hoc* and *fine-tuned*. *Post-hoc* methods directly modify learned representations without altering the parameters of the backbone model. In contrast, *fine-tuned* approaches require access to the model and enforce erasure by finetuning its parameters. A summary of concept-erasure methods can be found in Table 1.

**Linearly Guarded Erasure**: Early concept removal methods were linear, relying on identifying a direction associated with the unwanted concept and subtracting it from the representations [5, 15]. Iterative Nullspace Projection (INLP) improves on this by repeatedly projecting representations onto the nullspace of linear classifiers trained to predict the sensitive attribute, gradually removing its influence [26]. Relaxed Linear Adversarial Concept Erasure (R-LACE) formulates the task as a minimax game, where an adversarial classifier attempts to recover the concept being erased [27].

Table 1: Characteristics of Existing Concept Erasure Methods.

| Method | UnSupervised | Supervised | Erasure Model |
|--------|--------------|------------|---------------|
| INLP [26] | Post-hoc | Post-hoc | Linear/Iterative |
| kSAL/SAL [30] | Post-hoc | Post-hoc | Nonlinear |
| AdS [1] | - | Fine-tuning | Nonlinear |
| FaRM [6] | Post-hoc | Fine-tuning | Nonlinear |
| KRaM [2] | Post-hoc | Post-hoc | Nonlinear |
| Obliviator [Ours] | Post-hoc | Post-hoc | Nonlinear/Iterative |

LEAst-squares Concept Erasure (LEACE) takes a least-squares approach, yielding an oblique projection for linear concept removal [3]. Spectral Attribute removaL (SAL) uses matrix decomposition to project representations into directions with lower covariance with the sensitive labels [30]. However, none of these methods is robust against nonlinear adversaries.

**Nonlinearly Guarded Erasure**: Kernel-based techniques have been used to extend linearly guarded erasure methods to nonlinear ones. For example, Kernelized Concept Erasure (KCE) and kSAL are kernelized versions of R-LACE and SAL, respectively. However, these methods only protects the unwanted attributes against a specific class of adversaries and fail to ensure complete nonlinearly guarded erasure [28, 30]. Adversarial training has emerged as a common technique for concept erasure [12, 20, 32, 1]. In this method, an adversarial network is trained to predict the sensitive attribute, while the primary model learns to suppress it and preserve task-relevant information. A representative example is Adversarial Scrubbing (AdS) [1]. However, adversarial removal is often incomplete, and such models can be challenging to train effectively [13]. Another line of work for nonlinear concept erasure is based on rate-distortion maximization, as in FaRM [6], and its kernelized variant, KRaM [7]. Notably, both AdS and FaRM require fine-tuning PLMs. However, we show that even these approaches are still vulnerable to nonlinear adversaries and protected attributes can still be partially recovered using a proper classifier.

**Obliviator** addresses the gap in nonlinear guardedness by directly targeting the nonlinear statistical dependency between unwanted attributes and learned representations. We adopt a functional perspective to formulate this dependency. Similar to prior work [10, 19], we use the Hilbert-Schmidt Independence Criterion (HSIC) as a proxy to minimize this dependency. We propose a two-step iterative optimization that minimizes this quantity while enabling more utility-preserving erasure.

## 3 Statistical Dependence Measures

In this section, we explain how a linear statistical test can be derived from linear functions. We then extend this approach to a nonlinear test with functions from RKHS, leading to the definition of HSIC that captures nonlinear dependence of random variables and has a closed-form solution.

**Notation**: We denote scalars and functions using lowercase letters, e.g., $n$ and $f$. For deterministic vectors boldface lowercase letters, e.g., $\boldsymbol{x}, \boldsymbol{y}$ is used. Both scalar-valued and multidimensional random variables (RV)s are denoted by regular upper case letters, e.g., $X, Y$. We denote deterministic matrices by boldface upper case letters, e.g., $\boldsymbol{K}, \boldsymbol{L}$. Sets are showed by calligraphic letters, e.g., $\mathcal{F}, \mathcal{Z}$.

### 3.1 Dependence via Witness Functions

**Linear Statistical Dependence**: Let $X \in \mathbb{R}^n$ and $Y \in \mathbb{R}^m$ be two RVs with the joint distribution $p_{X,Y}$. To design a linear statistical test, we can look for subspaces $\boldsymbol{u}$ and $\boldsymbol{v}$ that maximize the correlation of the projected RVs. Therefore, we can write:

$$\sup_{\boldsymbol{v}} \sup_{\boldsymbol{u}} \mathbb{E}\left[\boldsymbol{v}^T \bar{Y} \bar{X}^T \boldsymbol{u}\right] = \boldsymbol{v}^T \boldsymbol{C}_{yx} \boldsymbol{u}, \quad \text{s.t} \quad \|\boldsymbol{u}\|_2 = \|\boldsymbol{v}\|_2 = 1 \tag{1}$$

where $\boldsymbol{C}_{yx} = \mathbb{E}[\bar{Y}\bar{X}^T]$ is the cross-covariance matrix, and $\bar{X} = X - \mathbb{E}[X]$ shows a centered RV. This optimization is bounded by the largest singular value of $\boldsymbol{C}_{yx}$, with $\boldsymbol{u}$ and $\boldsymbol{v}$ as its right and left singular vectors. Equivalently, one may consider all non-zero singular values of the cross-covariance matrix and introduce their sum of square as a Linear Independence Metric (LIM):

$$\text{LIM}(X, Y) = \sum_{i=1}^{r_c} \sigma_i^2 = \|\boldsymbol{C}_{yx}\|_F^2 \tag{2}$$

where $\|.\|_F$ denotes Frobenius norm and $r_c$ is the rank of $\boldsymbol{C}_{yx}$. This form of measuring dependence is similar to SAL [30] but fails to capture all modes of dependency when the dependence between $X$ and $Y$ is non-linear.

**Nonlinear Statistical Dependence**: We now extend the linear test, by replacing $\boldsymbol{u}$ and $\boldsymbol{v}$ with nonlinear functions. Let $\mathcal{F} : \boldsymbol{x} \to \phi(\boldsymbol{x})$ and $\mathcal{G} : \boldsymbol{y} \to \psi(\boldsymbol{y})$ be RKHSs where $\phi(.)$ and $\psi(.)$ are the corresponding feature maps. Also, let $\langle ., . \rangle_{\mathcal{H}}$ be an inner product for the function space $\mathcal{H}$. For a nonlinear statistical test, we look for a pair of functions $f \in \mathcal{F} : \mathbb{R}^n \to \mathbb{R}$ and $g \in \mathcal{G} : \mathbb{R}^m \to \mathbb{R}$ that maximize the following objective:

$$\sup_f \sup_g \mathbb{E}[\bar{g}(Y)\bar{f}(X)] = \langle g, \mathbb{C}\mathrm{ov}_{yx} f \rangle_{\mathcal{G}} \quad \text{s.t} \quad \|g\|_{\mathcal{G}} = \|f\|_{\mathcal{F}} = 1 \tag{3}$$

Here, $\bar{f}(X) := f(X) - \mathbb{E}[f(X)]$, and $\mathbb{C}\mathrm{ov}_{yx} = \mathbb{E}[\bar{\psi}(Y) \otimes \bar{\phi}(X)] : \mathcal{F} \to \mathcal{G}$ is the cross-covariance operator, where $\otimes$ denotes the tensor product, which extends the outer product to function spaces. The functions $f$ and $g$, act as *witness functions* that aim to transform the dependency between the input variables $X$ and $Y$ into an approximately linear relationship in the transformed spaces $f(X)$ and $g(Y)$. The unit norm constraint on these functions encourages smoothness. Since the underlying data distribution is unknown in practice, this constraint is essential for reliably estimating the objective from finite samples. Without such a constraint, it is possible to find very complex functions that produce a nonzero objective even when $X$ and $Y$ are independent.[1] To ensure the test captures a meaningful dependency from finite samples, we assume that any dependency between $X$ and $Y$ exhibits a smooth structure. This smoothness can also be influenced by the choice of the underlying feature map in the RKHS.

## 3.2 Hilbert-Schmidt Independence Criterion

The objective in (3) has a closed-form solution, where $f$ and $g$ are the right and left singular functions of the cross-covariance operator [17]. Similar to the case of linear dependence, HSIC considers the sum of square of all singular values of the covariance operator as a metric for independence:

$$\mathrm{HSIC}\big(\mathcal{F}, \mathcal{G}, p_{X,Y}\big) := \|\mathbb{C}\mathrm{ov}_{yx}\|_{HS}^2 = \sum_{i=1}^{r_c} \sigma_i^2 \tag{4}$$

Here, $\| \cdot \|_{HS}$ is the Hilbert-Schmidt norm, which generalizes the Frobenius norm to function spaces, and $r_c$ is the rank of the cross-covariance operator.

**Empirical Estimation of HSIC**: Let $\phi(\boldsymbol{x}) = k(., \boldsymbol{x})$ and $\psi(\boldsymbol{y}) = l(., \boldsymbol{y})$ be the corresponding kernels of $\mathcal{F}$ and $\mathcal{G}$. Given a set of observations $\mathcal{D} := \{(\boldsymbol{x}_i, \boldsymbol{y}_i)\}_{i=1}^n$ a biased estimator of HSIC is:

$$\widehat{\mathrm{HSIC}}(\mathcal{F}, \mathcal{G}, \mathcal{D}) = \frac{1}{n^2}\mathrm{trace}(\boldsymbol{K}\boldsymbol{H}\boldsymbol{L}\boldsymbol{H}) \tag{5}$$

where $\boldsymbol{K}$ and $\boldsymbol{L}$ are Kernel matrices for $\boldsymbol{x}$ and $\boldsymbol{y}$. $\boldsymbol{H} = \boldsymbol{I} - \frac{1}{n}\boldsymbol{1}_n\boldsymbol{1}_n^T$ is the centering matrix.

## 3.3 Independence From a Functional Perspective

Under the assumption that the corresponding kernels of $\mathcal{F}$ and $\mathcal{G}$ are characteristic, $X$ and $Y$ are statistically independent if no pair of witness functions can be found that yield a nonzero objective in Equation (3), or equivalently: $\mathrm{HSIC}(\mathcal{F}, \mathcal{G}, p_{X,Y}) = 0$ [18]. This principle distinguishes our method from prior kernel-based approaches, namely kSAL and KCE. These methods operate under the assumption that mapping a representation to RKHS and performing linear erasure in that space is sufficient for nonlinear guardedness. However, this only protects against linear adversaries within that specific feature space, remaining vulnerable to adversaries that exploit nonlinear patterns in that same space. This approach would guarantee statistical independence only if the RKHS feature map transforms representation to an approximately Gaussian distribution [29].

In contrast, Obliviator solves a more challenging problem: it seeks a representation $\varepsilon(X)$ where unwanted attribute $S$ remains undetectable even if transformed by a subsequent, adversarially chosen feature map $\phi(\cdot)$. This formulation results in a cascading kernel problem that lacks a closed-form solution and is more difficult to optimize. Mathematically, this is equivalent to finding a representation such that no pair of functions, $f$ and $g$, from a characteristic RKHS can be found to linearize the relationship between $f(\varepsilon(X))$ and $g(S)$ :

$$\inf_{\theta} \sup_g \sup_f \mathbb{E}\big[\bar{g}(S)\, \bar{f}\left(\varepsilon(\boldsymbol{\theta}; X)\right)\big] \tag{6}$$

where we assumed guarding function $\varepsilon$ is parameterized by $\boldsymbol{\theta}$. Thus, for the transformed RV $Z_\theta = \varepsilon(X)$ we have $\mathrm{HSIC}\big(\mathcal{F}, \mathcal{G}, p_{Z_\theta, S}\big) = 0$ and, therefore $Z_\theta \perp\!\!\!\perp S$.

---

[1]In extreme cases, such functions can be constructed using sharply peaked forms, such as the Dirac delta.

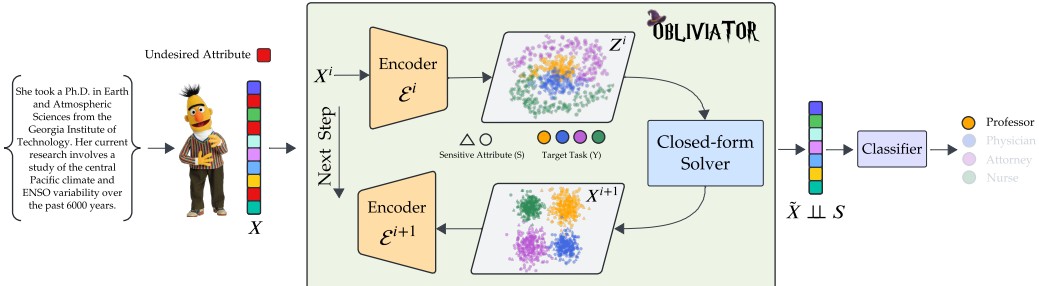

Figure 2: **Overview**. Obliviator operates with two-step iterations: 1) Imposing Independence via RKHS: An encoder is trained with a multi-objective loss (8) to reduce statistical dependence on the unwanted attribute while preserving task-relevant information. 2) RKHS Disentanglement: Representations from the previous step are refined using functions derived from a constrained optimization in RKHS (11). This refinement enhances the feature space's alignment with the target attribute, which facilitates the encoder's training in the next iteration. (Image source: BERT)

## 4    Obliviator: Non-Linearly Guarded Erasure

Obliviator solves (6), a challenging nested optimization problem. This formulation, however, lacks an explicit term to enforce the preservation of task-relevant information during erasure. RKHS witness functions make the adversary search tractable and, by a similar logic, the observability of the task-relevant information by them can be utilized as a proxy for information preservation. To formalize this, let $X$ be the original representation, with $S$ and $Y$ denoting the unwanted attribute and target task, respectively. Assume $\mathcal{F}$, $\mathcal{G}$, and $\mathcal{H}$ are the RKHSs from which we draw adversary and witness functions. Recalling our guarding function as $Z_\theta = \varepsilon(\boldsymbol{\theta}; X)$, we can write:

$$\inf_{\boldsymbol{\theta}} \quad \text{HSIC}(\mathcal{F}, \mathcal{G}, p_{Z_\theta, S}) - \text{HSIC}(\mathcal{F}, \mathcal{H}, p_{Z_\theta, Y}) \tag{7}$$

where we transformed the nested optimization (6) into a simpler bi-objective form using RKHS. This problem still lacks a closed-form solution and thus requires non-convex optimization. Due to its two competing objectives, single-shot optimization is likely to converge to a poor solution. We adopt an iterative approach to solve it which provides intermediate representations, offering more leverage to preserve utility during erasure. Moreover, we employ witness functions to reshape these intermediate representations such that the components of information independent of the unwanted attribute become more accessible for the encoder in the next iteration (see Figure 2 for an illustration).

> **Obliviator Two-step Process:**
> - *Imposing Independence via RKHS*: Utilizing RKHS, an encoder is trained to minimize the statistical dependence of the representation on unwanted attributes, while preserving task-relevant information by keeping it observable via witness functions.(§4.1).
> - *RKHS Disentanglement*: Solves a constrained optimization in RKHS to find functions that improve alignment of representation with utility, while avoiding such alignment with the unwanted attribute to facilitate the optimization in next iteration.(§4.2).

### 4.1   Encoder : Imposing Independence via RKHS

In our iterative method, we train a sequence of distinct encoders, where their inputs and outputs serve as intermediate RVs. Let $X^i$ be the input to the encoder at iteration $i$, and $Z_\theta^i = \varepsilon^i(\boldsymbol{\theta}^i; X^i)$ its output. Assume $X$, $Y$ and $S$ denote the RVs for the initial representation, target task, and unwanted attribute, respectively (see Figure 2). Our objective is to use witness functions as a proxy to impose the visibility of information about $Y$, $X$ and $X^i$ within $Z^i$, while minimizing this visibility for $S$ to achieve statistical independence. Recalling the objective in (7) and empirical estimation of HSIC from (5) we can write:

$$\inf_{\boldsymbol{\theta}^i} \quad \frac{1}{n^2}\text{trace}\Big(\boldsymbol{K}_{z_\theta^i}\boldsymbol{H}\big(\boldsymbol{K}_s - \tau_x\boldsymbol{K}_x - \tau_{x^i}\boldsymbol{K}_{x^i} - \tau_y\boldsymbol{K}_y\big)\boldsymbol{H}\Big) \tag{8}$$

where $\boldsymbol{K}_\bullet \in \mathbb{R}^{n\times n}$ denotes the kernel matrix for the corresponding variable. Compared to (7), we introduce two modifications: 1) $\tau_\bullet$ are hyperparameters that weight the importance of each objective.

2) $X$ and $X^i$ are auxiliary RVs included to enforce utility preservation. These are particularly useful when target-task labels $(Y)$ are unavailable.

A natural question, however, is why $X$ and $X^i$ are necessary when $Y$ is available. Recall from (3) that HSIC aggregates multiple "modes of visibility" (singular values) into a single scalar. Due to the competition between objectives, the optimization, which only sees this aggregate sum, is likely to discard weaker modes. The critical insight is that as information disentanglement evolves during optimization, the same mode of dependency is often weighted differently when measured relative to each RV. For instance, a specific mode of task-relevant information might be weakly expressed relative to $Y$ but remain more apparent relative to $X$ or $X^i$. Therefore, including $X$ and $X^i$ increases the likelihood that these modes are preserved during optimization.

## 4.2  Eigen Value Problem : RKHS Disentanglement

As discussed, the evolution of information disentanglement affects the visibility of dependency modes relative to each RV, a result of the competition between objectives in optimization. To counteract this, we introduce an intermediate step that utilizes witness functions to re-align the representation, making task-relevant information more accessible. Recalling the logic from (3) and (8), we seek a function $f \in \mathcal{F}$ that maximizes the correlation of $Z^i_\theta$ with the corresponding witness functions $g_\mathcal{I}$ in $\mathcal{G}_\mathcal{I}$ for $X, X^i$, and $Y$ (where $\mathcal{I} = \{x^i, x, y\}$). However, the search for $f$ must be constrained to avoid modifying the visibility of $S$, which would effectively counteract the minimization from the previous iteration. To formalize this, we solve the following constrained optimization:

$$
\sup_f \; \sup_{\{g_\mathcal{I}\}} \quad \mathbb{E}[\bar{g}_{x^i}(X^i)\bar{f}(Z^i)]^2 + \tau_x \, \mathbb{E}[\bar{g}_x(X)\bar{f}(Z^i)]^2 + \tau_y \, \mathbb{E}[\bar{g}_y(Y)\bar{f}(Z^i)]^2
$$
$$
\text{s.t.} \quad \sup_{g_s} \mathbb{E}[\bar{g}_s(S)\bar{f}(Z^i)] = 0, \quad \|g_\mathcal{I}\|_{\mathcal{G}_\mathcal{I}} = \|f\|_\mathcal{F} = \|g_s\|_{\mathcal{G}_s} = 1 \tag{9}
$$

Here, $\tau_\bullet$ are hyperparameters that weight the importance of each objective. Let $\boldsymbol{K}_\bullet = \boldsymbol{L}_\bullet \boldsymbol{L}_\bullet^\top$ be a full-rank factorization of the kernel matrix for the corresponding RV. This $\boldsymbol{L}_\bullet$ can be viewed as a finite-dimensional feature map. Using this map, the empirical estimation of an objective term, for example between $Z^i$ and $Y$, can be written as:

$$
\mathbb{E}[\bar{g}_y(Y)\bar{f}(Z^i)] \approx \frac{1}{n}\sum_{j=1}^{n} \bar{g}_y(\boldsymbol{y}_j)\bar{f}(\boldsymbol{z}_j^i) = \frac{1}{n}\boldsymbol{u}_y^T \boldsymbol{L}_y^T \boldsymbol{H} \boldsymbol{L}_{z^i} \boldsymbol{v} = \boldsymbol{u}_y^T \widehat{\boldsymbol{C}}_{yz^i} \boldsymbol{v} \tag{10}
$$

Here, $\boldsymbol{u}_y$ and $\boldsymbol{v}$ are the equivalence of $g_y$ and $f$ in the finite dimensional feature map. The optimal functions for (9) can then be empirically estimated by solving the following eigenvalue problem :

$$
\boldsymbol{Q}^T\Big(\widehat{\boldsymbol{C}}_{x^iz^i}^T \widehat{\boldsymbol{C}}_{x^iz^i} + \tau_y\, \widehat{\boldsymbol{C}}_{yz^i}^T \widehat{\boldsymbol{C}}_{yz^i} + \tau_x\, \widehat{\boldsymbol{C}}_{xz^i}^T \widehat{\boldsymbol{C}}_{xz^i}\Big)\boldsymbol{Q}\boldsymbol{v} = \lambda\boldsymbol{v} \tag{11}
$$

where $\boldsymbol{Q}$ is an orthonormal basis for $\text{Null}(\widehat{\boldsymbol{C}}_{sz^i})$. Given an eigenvalue threshold, we select $m$ eigenvectors from (11) that correspond to this threshold. Let $\boldsymbol{V} = [\boldsymbol{v}_1, \ldots, \boldsymbol{v}_m]$ denotes the selected eigen vectors, encoded representation for the next iteration is then given by:

$$
\boldsymbol{X}^{i+1} = \boldsymbol{L}_{z^i}\boldsymbol{Q}\boldsymbol{V} \tag{12}
$$

We defer the derivation and practical implementation for Obliviator to Appendix.

## 5  Experiments

**Evaluated PLMs:**  We consider four models: BERT(`bert-base-uncased`)[11], GPT-2 [23], DeepSeek-LLM-7B-Chat [9], and Llama-3.2-1B-Instruct [16]. For BERT, we use the final hidden state of the `[CLS]` token as the text representation. For DeepMoji [14] we use the setup as in [26, 6]. For the other models, we apply mean pooling to their token embeddings. The embedding dimensions for DeepMoji, BERT, GPT-2, DeepSeek and LLaMa are 300, 768, 768, 4096, and 2048 respectively.

**Evaluated Setup:**  Obliviator can utilize target task labels $(Y)$, a setup we term *supervised* erasure. In the *unsupervised* setup, these $Y$ labels are unavailable. PLM representations are kept *frozen*, with the exception of BERT, for which we also consider a *fine-tuned* case where the model's weights are modified to improve disentanglement with respect to $Y$. We use three datasets in our experiments: BIAS IN BIOS [8] (Y: Profession, S: Gender), DIAL-SENTIMENT [4] (Y: Sentiment, S: Race), and

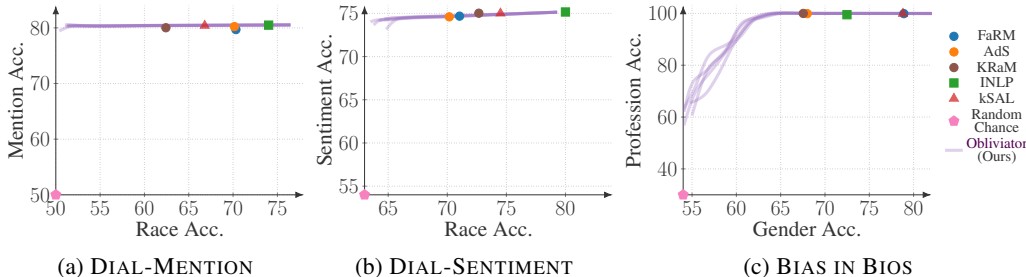

(a) DIAL-MENTION      (b) DIAL-SENTIMENT      (c) BIAS IN BIOS

Figure 3: **Finetuned+Supervised Erasure** : Comparison of Obliviator with baselines for fine-tuned representations. Obliviator leverages $Y$ labels during the erasure, a scheme which we refer to as supervised erasure."

DIAL-MENTION [4] (Y: Mention, S: Race). We compare Obliviator against INLP, AdS, kSAL, FaRM, and KRaM as baselines.

**Utility-Erasure Trade-off:** While concept erasure methods are typically evaluated on their final-stage, such evaluations overlook how competing objectives interact throughout the erasure process. Studying the utility-erasure trade-off provides deeper insight into this interaction, offering a more comprehensive assessment of a method's performance and its optimization. To construct these trade-off curves, we evaluate utility and leakage at each iteration. We measure attribute leakage by probing for the unwanted concept with various nonlinear adversaries and reporting their highest accuracy. We continue the erasure process until the probe accuracy for the unwanted attribute become close to *random chance accuracy*, which is the majority class percentage and shown as the origin of all trade-off plots. We compute trade-off curves from five independent runs.

> **Questions Investigated in Experiments:**
>
> §5.1 *How do different experimental scenarios affect utility-erasure trade-off?* Here, we analyze the effect of the erasure scheme (supervised vs. unsupervised), the initial representation (frozen vs. fine-tuned), and different datasets in BERT, a common benchmark that has been evaluated by many erasure methods.
>
> §5.2 *How does Obliviator generalize across different language models?* Here, our goal is to investigate whether the better disentanglement of representations learned by more capable language models is reflected in the utility-erasure trade-off.
>
> §5.3 *How does biased sampling or data skewness affect the erasure?* Here, we use another common benchmark in concept erasure, a controlled setup that modifies the proportion of the unwanted attribute, to investigate the resulting trade-off with the Deepmoji model.
>
> §5.4 *How do erasure schemes on different PLMs affect fairness metrics?* Here, we explore the effect of erasure on downstream fairness under these conditions.

## 5.1 How does different experimental scenarios affect utility-erasure trade-off ?

In this section, we compare Obliviator against baselines to examine the effects of the erasure scheme (supervised vs. unsupervised), the initial representation (frozen vs. fine-tuned), and different datasets mentioned above. We use the BERT model for this analysis, as it is a common benchmark on which many erasure methods have been evaluated using these datasets. However, the utility-erasure trade-off in these contexts has not been previously explored. We consider all four combinations of erasure scheme and representation, namely : 1)*Finetuned+Supervised* (Figure 3), 2)*Frozen+Unsupervised* (Figure 4), 3)*Frozen+Supervised* (Figure 5), 4)*Finetuned+Unsupervised* (Figure 5). First, we study the initial two scenarios, which are designed to assess the impact of $Y$ label availability, either for fine-tuning or during the erasure process. The final two scenarios then investigate how the visibility of $Y$ information affects the erasure process.

*Erasure Performance:* Considering Figure 3 and Figure 4, Obliviator achieves nonlinear guardedness, capturing the full trade-off curve across all scenarios and datasets. This outcome is expected, as our method directly minimizes HSIC $\to 0$, a condition equivalent to independence from a functional perspective. In contrast, existing methods fail to achieve full erasure, particularly for fine-tuned representations. Notably, in Figure 3, the gap between full erasure (i.e., random chance) and the best-performing baseline is $12\%$, $13\%$, and $14\%$ for DIAL-MENTION, DIAL-SENTIMENT, and BIAS

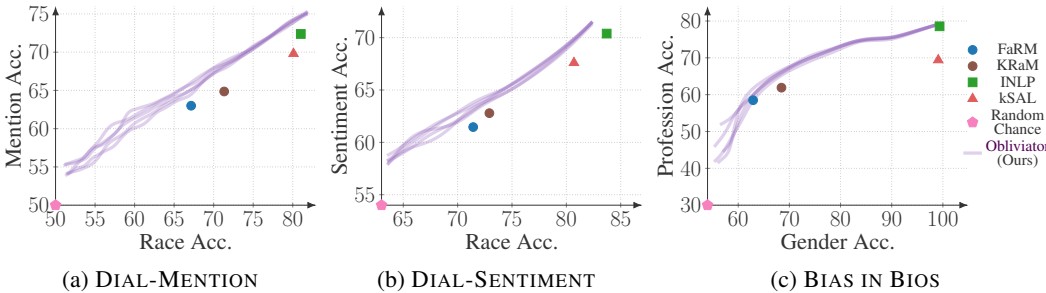

(a) DIAL-MENTION  (b) DIAL-SENTIMENT  (c) BIAS IN BIOS

Figure 4: **Frozen+Unsupervised Erasure** : Comparison of Obliviator and baselines with frozen representations. In unsupervised erasure, we implicitly observe $Y$ information from $X$ and $X^i$ and thereby we observe a more noticeable trade-off compared to Figure 3.

IN BIOS, respectively. Similar gaps of $12\%$, $9\%$, and $8\%$ are observed in Figure 4. Obliviator consistently outperforms all baselines, achieving higher Y accuracy at every level of S accuracy across all scenarios and datasets. This result stems from our algorithm's two core components: the utility-preserving loss in (8) and the disentanglement process in (11). Notably, utilizing RKHS to formulate these two components in closed-form is a key contributor to the consistency of our optimization.

***Cost of Erasure:*** Comparing the trade-off curves for each dataset across the different setups in Figure 3 and Figure 4, we observe a more noticeable trade-off in the *frozen+unsupervised* scenario. This is consistent with our discussion in Section 4.1. The key difference is that $Y$ provides an explicit proxy for the task-relevant modes. In contrast, $X$ and $X^i$ are implicit proxies; the optimization's priority for the task-relevant modes depends on how those modes are weighted relative to all other information encoded within these RVs. Therefore, preserving $Y$-relevant information is more likely in the supervised erasure setup. This distinction is particularly evident in DIAL-MENTION and DIAL-SENTIMENT, where Obliviator preserves mention/sentiment information with minimal utility loss while successfully erasing race-related information.

***Effect of $Y$ Visibility on Erasure:*** To distinguish the effect of the $Y$ label on *supervised* erasure from its effect via *fine-tuning*, we examine two key scenarios: *Frozen+Supervised* and *Finetuned+Unsupervised*. First, we analyze *Frozen+Supervised* for DIAL-MENTION (Figure 5a). The high utility preservation confirms that supervised erasure alone is highly effective, strengthening our hypothesis from Section 4.1 that an explicit proxy for $Y$-relevant modes is more likely to preserve that information. Interestingly, the *Finetuned+Unsupervised* setup for DIAL-

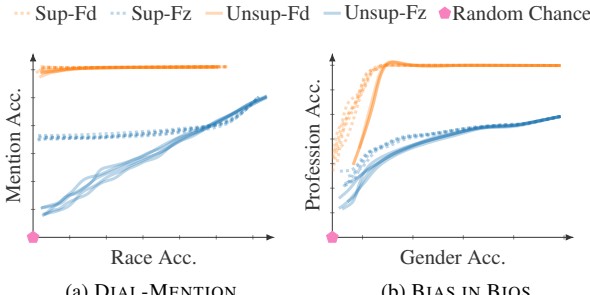

Figure 5: Supervised and unsupervised erasure on fine-tuned and frozen representations. (Sup: Supervised, Unsup: Unsupervised, Fd: Finetuned, and Fz: Frozen.)

MENTION also preserves utility well. This indicates that fine-tuning strengthens the visibility of $Y$-relevant information when measured implicitly using $X$ and $X^i$, giving this information a higher priority in the optimization.

However, the behavior differs for BIAS IN BIOS. As shown in Figure 5b, the gap between erasure schemes on the frozen representation is less noticeable than DIAL-MENTION. This could be due to the nature of the $Y$ labels: BIAS IN BIOS has 28 classes, so it is likely that task-relevant modes have higher weights when measured by witness functions from $X$ and $X^i$. In contrast, for DIAL-MENTION (binary $Y$), the signal is weaker, making the implicit proxies $X$ and $X^i$ less practical compared to an explicit $Y$ proxy for preserving the information. For *Finetuned+Unsupervised* erasure in BIAS IN BIOS, we again observe that preserving $Y$ information is possible up to a point, implicitly through $X$ and $X^i$, for a similar reason as in DIAL-MENTION. However, in both the finetuned and frozen representations, we observe that supervised erasure preserves information much better. This can be attributed to the erasure process itself: as the representation evolves and becomes more disentangled, the visibility of $Y$ information through the implicit proxies ($X$ and $X^i$) may also evolve and weaken.

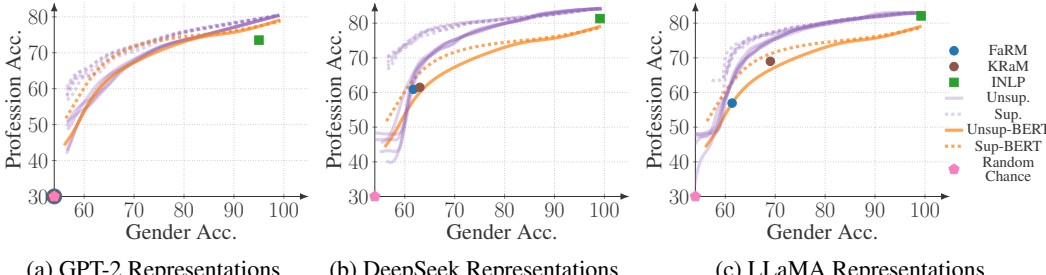

|  | (a) GPT-2 Representations | (b) DeepSeek Representations | (c) LLaMA Representations |

Figure 6: **Erasure Across Different PLMs Compared to BERT.** The figure shows *supervised* and *unsupervised* erasure using frozen representations from GPT-2, DeepSeek, and LLaMa on BIAS IN BIOS.

Therefore, having an explicit proxy for $Y$ information becomes more valuable as the optimization proceeds.

> **Takeaway 1.** In supervised erasure, Obliviator utilizes an explicit term to observe $Y$-relevant information via witness functions ((8) and (11)). This provides a direct optimization signal that is more likely to preserve utility, especially in the final stages where erasure targets more entangled information.
>
> **Takeaway 2.** In unsupervised erasure, the initial visibility of $Y$-relevant information affects its priority in the competition of objectives when measured via the implicit proxies ($X$ and $X^i$), determining its preservation during erasure.

### 5.2 How does Obliviator generalize across different language models?

Here, we examine the generalizability of Obliviator across different PLMs. We hypothesize that more capable models should learn representations that are better disentangled. Consequently, a robust erasure approach should yield improved trade-offs when applied to them. To explore this, we use *frozen* BIAS IN BIOS representations from GPT-2, LLaMa, and DeepSeek. Figure 6 shows:

***GPT-2 Representations***: The trade-off profile of GPT-2 closely matches that of BERT in both erasure schemes, an expected result given their comparable model complexity. However, the performance of several baselines, including INLP, FaRM, and KRaM, degrades on GPT-2. Notably, FaRM and KRaM drop to near random-chance accuracy on the target task.

***LLaMa Representations***: Compared to BERT, we observe clear improvements in both erasure schemes. As LLaMa provides representations with better initial disentanglement of $Y$ and $S$, Obliviator leverages this to achieve a more utility-preserving erasure. This result demonstrates the generalizability of our method and the effectiveness of its optimization approach. The performance of FaRM and INLP remains largely unchanged from BERT, and while KRaM shows some improvement, it still fails to achieve full erasure. Notably, Obliviator consistently outperforms all baselines.

***DeepSeek Representations***: In the *unsupervised* erasure, DeepSeek representations yield a performance improvement over BERT similar to that observed with LLaMa. Among the baselines, KRaM's erasure performance improves compared to its results on LLaMa representations, while FaRM's behavior remains roughly unchanged. For target task accuracy, KRaM's performance degrades, whereas FaRM shows some improvement. Obliviator consistently outperforms all baselines. Interestingly, in *supervised* erasure, we observe a significant improvement in the trade-off profile, with only mild degradation in target accuracy as erasure completes. This demonstrates that achieving full erasure on the BIAS IN BIOS dataset while maintaining substantial target task information is possible. This strongly suggests that the observed utility-erasure trade-off is a diagnostic of the specific model's learned dependencies, not necessarily an inherent real-world link.

> **Takeaway .** With better disentangled representations learned by more capable PLMs, Obliviator's erasure becomes more utility-preserving, indicating its strong generalizability.

### 5.3 How does biased sampling or data skewness affect the erasure?

In this section, we explore the effect of biased sampling on erasure, using a controlled setup recognized as a classic benchmark in concept erasure [13, 26, 6]. Following this setup, we use the DIAL-

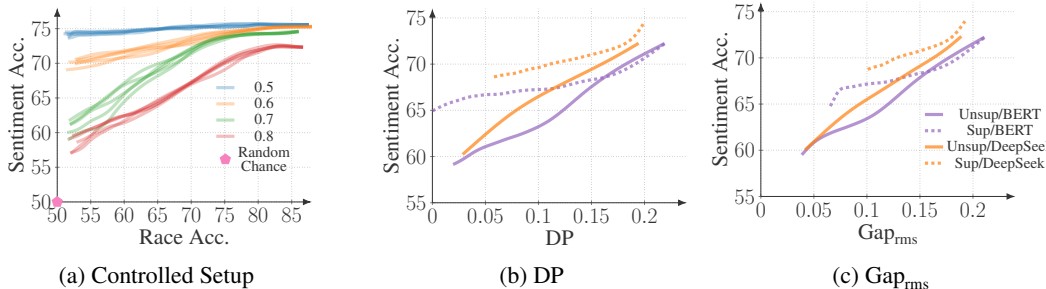

| (a) Controlled Setup | (b) DP | (c) Gap$_{\text{rms}}$ |

Figure 7: (**a**). Unsupervised erasure for DeepMoji representations on DIAL-SENTIMENT, plotted against varying levels of unwanted attribute disproportion. (**b-c**). Demographic Parity (DP) and Gap$_{\text{rms}}$ across different erasure scheme and PLMs in DIAL-SENTIMENT.

SENTIMENT dataset, DeepMoji representations, and an unsupervised erasure scheme. For biased sampling, we modified the proportion of the unwanted attribute (Race) within each target task class (Sentiment). For instance, in the 80% split, the "happy" sentiment class is composed of 80% African-American English (AAE) and 20% Standard American English (SAE), while the "sad" sentiment class is composed of 20% AAE and 80% SAE. We hypothesized that increasing this imbalance would worsen the trade-off, as a skewed sample provides a biased estimate of the true statistical dependence (HSIC) and makes finding a utility-preserving solution more difficult. Figure 7a confirms this hypothesis, as we observe the trade-off becomes more noticeable as the sampling skew increases. This highlights an inherent challenge for post-hoc erasure: the observed trade-off is highly dependent on how well the erasure dataset represents the true data distribution.

> **Takeaway .** Skewed sampling worsens the trade-off due to increased bias in estimation of HSIC.

### 5.4 How do erasure schemes on different PLMs affect fairness metrics?

Here, we investigate the effect of erasure on downstream fairness, speculating that imposing statistical independence w.r.t unwanted attribute will improve fairness metrics. We use Demographic Parity, $\text{DP} = \mathbb{E}_C[|\, p(\hat{Y} = C|S = 0) - p(\hat{Y} = C|S = 1)\,|]$, and $\text{Gap}_{\text{rms}} = \mathbb{E}_C[(p(\hat{Y} = C|Y = C, S = 0) - p(\hat{Y} = C|Y = C, S = 1))^2]^{1/2}$, where $\hat{Y}$ is the prediction of a model. With statistical independence w.r.t $S$, fairness metrics should improve and theoretically $\text{DP} \to 0$. We chose DIAL-SENTIMENT as our dataset, with DeepSeek and BERT as the two PLMs, as they encode entanglements differently. The results for DP and Gap$_{\text{rms}}$ are shown in Figure 7b and Figure 7c, respectively. Obliviator improves fairness metrics across erasure schemes, with the supervised scheme being more utility-preserving. Moreover, due to the better disentanglement of representations in DeepSeek, we observe a higher utility for any given level of unfairness compared to BERT.

> **Takeaway .** Obliviator aids fairness by imposing statistical independence w.r.t unwanted attributes.

## 6 Concluding Remarks

Concept erasure aims to remove unwanted attributes from representations while guarding against all adversaries. Existing methods fail to address this challenge, as they remain vulnerable to nonlinear adversaries. We propose Obliviator, which formulates erasure from a functional perspective to capture nonlinear statistical dependencies. We solve the resulting, challenging nested optimization with a two-step iterative approach. We investigate the effect of erasure more comprehensively by analyzing the full utility-erasure trade-off, a dynamic overlooked in the literature. We explored how different erasure schemes and the disentanglement of the original representation affect this trade-off. Our results show that Obliviator achieves state-of-the-art performance, provides robustness against nonlinear adversaries, and sets a strong benchmark for the utility-erasure trade-off. This performance remains consistent across various datasets and language models. The improved disentanglement of representations from more capable models is directly reflected in Obliviator's utility-erasure trade-off profiles, indicating its strong generalizability and effectiveness of its optimization approach.

**Acknowledgements:** This work was supported by the National Science Foundation (award #2147116). Any opinions, findings, conclusions, or recommendations expressed in this material are those of the authors and do not necessarily reflect the views of the NSF.

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

# Supplementary Material for Obliviator

This supplementary material provides additional details to support the main paper. It includes mathematical proofs, implementation specifics, ablation studies, and additional results to further illustrate and validate the effectiveness of Obliviator. The contents are organized as follows:

## A    Mathematical Proofs

### A.1    Cross-Covariance Operator

Let $\mathcal{F} : \boldsymbol{x} \mapsto \phi(\boldsymbol{x})$ and $\mathcal{G} : \boldsymbol{y} \mapsto \psi(\boldsymbol{y})$ denote reproducing kernel Hilbert spaces (RKHSs) with feature maps $\phi$ and $\psi$, respectively. We aim to find functions $f \in \mathcal{F}$ and $g \in \mathcal{G}$ that maximize the following objective:

$$\sup_{g \in \mathcal{G}} \sup_{f \in \mathcal{F}} \quad \mathbb{E}\left[\left(f(\boldsymbol{x}) - \mathbb{E}[f(\boldsymbol{x})]\right)\left(g(\boldsymbol{y}) - \mathbb{E}[g(\boldsymbol{y})]\right)\right] \quad s.t \quad \|f\|_{\mathcal{F}} = \|g\|_{\mathcal{G}} = 1 \tag{13}$$

Using the reproducing property of RKHSs, the centered evaluation of $f$ can be expressed as:

$$\begin{aligned}
\bar{f}(\boldsymbol{x}) &= \langle \phi(\boldsymbol{x}), f \rangle_{\mathcal{F}} - \mathbb{E}[\langle \phi(\boldsymbol{x}), f \rangle_{\mathcal{F}}] \\
&= \langle \phi(\boldsymbol{x}), f \rangle_{\mathcal{F}} - \langle \mathbb{E}[\phi(\boldsymbol{x})], f \rangle_{\mathcal{F}} \\
&= \langle \phi(\boldsymbol{x}) - \mathbb{E}[\phi(\boldsymbol{x})], f \rangle_{\mathcal{F}} \\
&= \langle \bar{\phi}(\boldsymbol{x}), f \rangle_{\mathcal{F}}
\end{aligned} \tag{14}$$

where $\bar{\phi}(\boldsymbol{x}) := \phi(\boldsymbol{x}) - \mathbb{E}[\phi(\boldsymbol{x})]$. A similar expression holds for $g(\boldsymbol{y})$. Substituting into the objective, we obtain:

$$\sup_{g \in \mathcal{G}} \sup_{f \in \mathcal{F}} \quad \mathbb{E}\left[\bar{f}(\boldsymbol{x})\bar{g}(\boldsymbol{x})\right] = \mathbb{E}\left[\langle \bar{\phi}(\boldsymbol{x}), f \rangle_{\mathcal{F}} \langle \bar{\psi}(\boldsymbol{y}), g \rangle_{\mathcal{G}}\right] \tag{15}$$

Recall that the tensor product $g \otimes f : \mathcal{F} \to \mathcal{G}$ is a rank-one operator defined by:

$$(g \otimes f)h := g\langle f, h \rangle_{\mathcal{F}}, \quad \text{for all } h \in \mathcal{F}. \tag{16}$$

Using this, the objective becomes:

$$
\begin{aligned}
\mathbb{E}\left[\langle\bar{\phi}(\boldsymbol{x}),f\rangle_{\mathcal{F}}\,\langle\bar{\psi}(\boldsymbol{y}),g\rangle_{\mathcal{G}}\right] &= \mathbb{E}\left[\langle\bar{\psi}(\boldsymbol{y})\,\langle\bar{\phi}(\boldsymbol{x}),f\rangle\,,g\rangle_{\mathcal{G}}\right] \\
&= \mathbb{E}\left[\langle\bar{\psi}(\boldsymbol{y})\otimes\bar{\phi}(\boldsymbol{x})\,f,g\rangle_{\mathcal{G}}\right] \\
&= \left\langle\mathbb{E}[\bar{\psi}(\boldsymbol{y})\otimes\bar{\phi}(\boldsymbol{x})]\,f,g\right\rangle_{\mathcal{G}} \\
&= \langle\mathbb{C}\mathrm{ov}_{yx}f,g\rangle_{\mathcal{G}} = \langle g,\mathbb{C}\mathrm{ov}_{yx}f\rangle_{\mathcal{G}}
\end{aligned}
\tag{17}
$$

where $\mathbb{C}\mathrm{ov}_{yx} := \mathbb{E}[\bar{\psi}(\boldsymbol{y})\otimes\bar{\phi}(\boldsymbol{x})]$ denotes the cross-covariance operator from $\mathcal{F}$ to $\mathcal{G}$.

***Remark.*** Strictly speaking, interchanging the expectation with the inner product requires justifying that $\bar{\psi}(\boldsymbol{y})\otimes\bar{\phi}(\boldsymbol{x})$ defines a Hilbert–Schmidt (HS) operator and that its expectation exists in the Hilbert space of HS operators. We provide a minimal justification in the following section. The derivation above is retained for its close resemblance to the finite-dimensional cross-covariance formulation.

## A.2 Hilbert–Schmidt Operators

Let $\mathcal{F}$ and $\mathcal{G}$ be separable Hilbert spaces and $\{q_i\}_{i=1}^{\infty}$ an orthonormal basis of $\mathcal{F}$. For a bounded operator $\mathscr{L} : \mathcal{F}\to\mathcal{G}$ the *Hilbert–Schmidt(HS)* norm is defined as

$$
\|\mathscr{L}\|_{\mathrm{HS}}^2 = \sum_{i=1}^{\infty}\|\mathscr{L}q_i\|_{\mathcal{G}}^2
\tag{18}
$$

If this series converges, $\mathscr{L}$ is called *Hilbert–Schmidt*. The set of Hilbert-Schmidt operators mapping from $\mathcal{F}$ to $\mathcal{G}$ is a Hilbert space denoted by $\mathrm{HS}(\mathcal{F},\mathcal{G})$ with the inner product

$$
\langle\mathscr{L},\mathscr{T}\rangle_{\mathrm{HS}} = \sum_{i=1}^{\infty}\langle\mathscr{L}q_i,\mathscr{T}q_i\rangle_{\mathcal{G}}
\tag{19}
$$

**Rank one tensor product is HS.** For $f \in \mathcal{F}$ and $g \in \mathcal{G}$:

$$
\begin{aligned}
\|g\otimes f\|_{\mathrm{HS}}^2 &= \sum_{i=1}^{\infty}\|g\langle f,q_i\rangle_{\mathcal{F}}\|_{\mathcal{G}}^2 \\
&= \|g\|_{\mathcal{G}}^2\sum_{i=1}^{\infty}|\langle f,q_i\rangle_{\mathcal{F}}|^2 \\
&= \|g\|_{\mathcal{G}}^2\|f\|_{\mathcal{F}}^2 < \infty
\end{aligned}
\tag{20}
$$

so $g \otimes f \in \mathrm{HS}(\mathcal{F},\mathcal{G})$.

**Lemma.** *For any $\mathscr{L} \in \mathrm{HS}(\mathcal{F},\mathcal{G})$, $f \in \mathcal{F}$, and $g \in \mathcal{G}$,*

$$
\langle\mathscr{L},\,g\otimes f\rangle_{\mathrm{HS}} = \langle g,\mathscr{L}f\rangle_{\mathcal{G}}
\tag{21}
$$

*Proof.* Choose an orthonormal basis of $\mathcal{F}$ that begins with the normalised vector $\tilde{f} := f/\|f\|_{\mathcal{F}}$, i.e. $\{\tilde{f}\}\cup\mathcal{B}_{\perp}$ with $\mathcal{B}_{\perp} := \{\tilde{f}_i^{\perp}\}_{i=1}^{\infty}$ :

$$
\begin{aligned}
\langle\mathscr{L},\,g\otimes f\rangle_{\mathrm{HS}} &= \langle\mathscr{L}\tilde{f},\,g\otimes f\,\tilde{f}\rangle_{\mathcal{G}} + \overbrace{\sum_{q_i\in\mathcal{B}_{\perp}}\langle\mathscr{L}q_i,\,g\otimes f\,q_i\rangle_{\mathcal{G}}}^{=0} \\
&= \frac{1}{\|f\|_{\mathcal{F}}}\langle\mathscr{L}f,\,g\,\langle f,\tilde{f}\rangle_{\mathcal{F}}\rangle_{\mathcal{G}} \\
&= \langle g,\mathscr{L}f\rangle_{\mathcal{G}}
\end{aligned}
\tag{22}
$$

The second term vanishes because $f$ is orthogonal to every $q_i\in\mathcal{B}_{\perp}$. $\qquad\square$

Using this lemma we have :

$$\langle \bar{\psi}(\boldsymbol{y}) \otimes \bar{\phi}(\boldsymbol{x}),\, g \otimes f \rangle_{\text{HS}} = \langle \bar{\phi}(\boldsymbol{x}), f \rangle_{\mathcal{F}} \langle \bar{\psi}(\boldsymbol{y}), g \rangle_{\mathcal{G}} \tag{23}$$

and

$$\mathbb{E}\big[\bar{f}(\boldsymbol{x})\bar{g}(\boldsymbol{y})\big] = \mathbb{E}\big[\langle \bar{\psi}(\boldsymbol{y}) \otimes \bar{\phi}(\boldsymbol{x}),\, g \otimes f \rangle_{\text{HS}}\big] \tag{24}$$

**Boundedness of the expectation functional.** Consider the following linear functional:

$$\mathscr{F} : \text{HS}(\mathcal{F}, \mathcal{G}) \to \mathbb{R}, \qquad \mathscr{F}(\mathscr{L}) = \mathbb{E}\big[\langle \bar{\psi}(\boldsymbol{y}) \otimes \bar{\phi}(\boldsymbol{x}),\, \mathscr{L} \rangle_{\text{HS}}\big] \tag{25}$$

Applying Jensen and then Cauchy–Schwarz inequalities:

$$\begin{aligned}
\big|\mathscr{F}(\mathscr{L})\big| &\leq \mathbb{E}\big[\big|\langle \bar{\psi}(\boldsymbol{y}) \otimes \bar{\phi}(\boldsymbol{x}),\, \mathscr{L} \rangle_{\text{HS}}\big|\big] \\
&\leq \mathbb{E}\big[\|\bar{\psi}(\boldsymbol{y}) \otimes \bar{\phi}(\boldsymbol{x})\|_{\text{HS}}\, \|\mathscr{L}\|_{\text{HS}}\big] \\
&\leq \|\mathscr{L}\|_{\text{HS}}\, \mathbb{E}\big[\|\bar{\phi}(\boldsymbol{x})\|_{\mathcal{F}}\, \|\bar{\psi}(\boldsymbol{y})\|_{\mathcal{G}}\big] \\
&\leq \|\mathscr{L}\|_{\text{HS}}\, \mathbb{E}\big[\sqrt{k(\boldsymbol{x}, \boldsymbol{x})\, l(\boldsymbol{y}, \boldsymbol{y})}\big] < \infty
\end{aligned} \tag{26}$$

so $\mathscr{F}$ is a bounded linear functional on $\text{HS}(\mathcal{F}, \mathcal{G})$.[2]

**Riesz representation and the covariance operator.** Recall that on any Hilbert space, the Riesz representation theorem states that every bounded linear functional can be expressed as an inner product with a unique element of that space. Applying this to the bounded functional $\mathscr{F}$ defined above, we obtain a unique operator $\mathbb{C}\text{ov}_{yx} \in \text{HS}(\mathcal{F}, \mathcal{G})$ satisfying:

$$\big\langle \mathbb{C}\text{ov}_{yx},\, \mathscr{L} \big\rangle_{\text{HS}} = \mathscr{F}(\mathscr{L}) \qquad \forall\, \mathscr{L} \in \text{HS}(\mathcal{F}, \mathcal{G}) \tag{27}$$

Taking $\mathscr{L} = g \otimes f$ recovers the cross-covariance identity used in the main text. We now turn to the empirical estimation of cross-covariance operator.

### A.3 Empirical Estimation of the Objective

Given a dataset $\mathcal{D} = \{(\boldsymbol{x}_i, \boldsymbol{y}_i)\}_{i=1}^{n}$, the empirical estimate of the cross-covariance operator is given by:

$$\widehat{\mathbb{C}\text{ov}}_{yx} = \frac{1}{n} \sum_{i=1}^{n} \bar{\psi}(\boldsymbol{y}_i) \otimes \bar{\phi}(\boldsymbol{x}_i). \tag{28}$$

Next, by examining the action of the tensor product operator in Equation (16), together with the unit-norm constraints on $f$ and $g$, we conclude[3]:

$$f = \sum_{i=1}^{n} \alpha_i \bar{\phi}(\boldsymbol{x}_i), \quad \text{and} \quad g = \sum_{i=1}^{n} \beta_i \bar{\psi}(\boldsymbol{y}_i) \tag{29}$$

Substituting this representation into the objective in Equation (17), we obtain:

$$\mathbb{E}\big[\bar{f}(\boldsymbol{x})\bar{g}(\boldsymbol{x})\big] \approx \frac{1}{n} \sum_{i=1}^{n} \sum_{j=1}^{n} \sum_{k=1}^{n} \alpha_j \beta_k \langle \bar{\psi}(\boldsymbol{y}_i), \bar{\psi}(\boldsymbol{y}_k) \rangle_{\mathcal{G}} \cdot \langle \bar{\phi}(\boldsymbol{x}_i), \bar{\phi}(\boldsymbol{x}_j) \rangle_{\mathcal{F}} \tag{30}$$

Letting $k$ and $l$ denote the kernels associated with $\mathcal{F}$ and $\mathcal{G}$, respectively, the expression simplifies to:

$$\mathbb{E}\big[\bar{f}(\boldsymbol{x})\bar{g}(\boldsymbol{x})\big] \approx \frac{1}{n} \sum_{i=1}^{n} \sum_{j=1}^{n} \sum_{k=1}^{n} \beta_k\, \bar{l}_{ik}\, \bar{k}_{ij}\, \alpha_j = \frac{1}{n} \boldsymbol{\beta}^{\top} \bar{\boldsymbol{L}} \bar{\boldsymbol{K}} \boldsymbol{\alpha} \tag{31}$$

where $\bar{\boldsymbol{K}}$ and $\bar{\boldsymbol{L}}$ are the centered kernel matrices corresponding to $k$ and $l$, respectively. Following the representation of $f$ and $g$ from Equation (29), the unit-norm constraint on $f$ can be expressed as:

$$\begin{aligned}
\|f\|_{\mathcal{F}}^2 &= \left\langle \sum_{i=1}^{n} \alpha_i \bar{\phi}(\boldsymbol{x}_i), \sum_{j=1}^{n} \alpha_j \bar{\phi}(\boldsymbol{x}_j) \right\rangle_{\mathcal{F}} \\
&= \sum_{i=1}^{n} \sum_{j=1}^{n} \alpha_i \alpha_j \big\langle \bar{\phi}(\boldsymbol{x}_i), \bar{\phi}(\boldsymbol{x}_j) \big\rangle_{\mathcal{F}} \\
&= \boldsymbol{\alpha}^{\top} \bar{\boldsymbol{K}} \boldsymbol{\alpha}
\end{aligned} \tag{32}$$

---

[2]We assume $k(\boldsymbol{x}, \boldsymbol{x})$ and $l(\boldsymbol{y}, \boldsymbol{y})$ have finite first moments.

[3]This follows directly from the Representer Theorem.

and likewise for $g$:

$$\|g\|_{\mathcal{G}}^2 = \boldsymbol{\beta}^\top \bar{\boldsymbol{L}} \boldsymbol{\beta} \tag{33}$$

Therefore, the empirical estimation of the objective reduces to the following constrained optimization problem:

$$\sup_{\boldsymbol{\alpha}} \sup_{\boldsymbol{\beta}} \frac{1}{n} \boldsymbol{\beta}^\top \bar{\boldsymbol{L}} \bar{\boldsymbol{K}} \boldsymbol{\alpha} \qquad \text{s.t.} \qquad \boldsymbol{\alpha}^\top \bar{\boldsymbol{K}} \boldsymbol{\alpha} = \boldsymbol{\beta}^\top \bar{\boldsymbol{L}} \boldsymbol{\beta} = 1 \tag{34}$$

Next, let $\bar{\boldsymbol{K}} = \boldsymbol{J}\boldsymbol{J}^\top$ and define $\boldsymbol{u} = \boldsymbol{J}^\top \boldsymbol{\alpha}$, and similarly let $\bar{\boldsymbol{L}} = \boldsymbol{D}\boldsymbol{D}^\top$ with $\boldsymbol{v} = \boldsymbol{D}^\top \boldsymbol{\beta}$. Then, the objective can be rewritten as:

$$\sup_{\boldsymbol{u}} \sup_{\boldsymbol{v}} \frac{1}{n} \boldsymbol{v}^\top \boldsymbol{D}^\top \boldsymbol{J} \boldsymbol{u} \qquad \text{s.t.} \qquad \boldsymbol{v}^\top \boldsymbol{v} = \boldsymbol{u}^\top \boldsymbol{u} = 1 \tag{35}$$

This is maximized by the largest singular value of $\boldsymbol{D}^\top \boldsymbol{J}$, i.e., it is upper bounded by $\sigma_{\max}(\boldsymbol{D}^\top \boldsymbol{J})$. Next, by considering the sum of squared singular values, we obtain:

$$\sum_{i=1}^n \sigma_i^2 = \frac{1}{n^2} \operatorname{tr}\left(\boldsymbol{J}^\top \boldsymbol{D}\boldsymbol{D}^\top \boldsymbol{J}\right) = \frac{1}{n^2} \operatorname{tr}\left(\boldsymbol{J}\boldsymbol{J}^\top \boldsymbol{D}\boldsymbol{D}^\top\right) = \frac{1}{n^2} \operatorname{tr}(\bar{\boldsymbol{K}}\bar{\boldsymbol{L}}) = \frac{1}{n^2} \operatorname{tr}(\boldsymbol{K}\boldsymbol{H}\boldsymbol{L}\boldsymbol{H}) \tag{36}$$

where we used the cyclic property of the trace and the centering matrix $\boldsymbol{H} = \boldsymbol{I} - \frac{1}{n}\boldsymbol{1}\boldsymbol{1}^\top$.

Finally, the squared *Hilbert–Schmidt* norm of the empirical cross-covariance operator reproduces the same scalar quantity:

$$
\begin{aligned}
\left\|\widehat{\mathbb{C}\mathrm{ov}}_{yx}\right\|_{\mathrm{HS}}^2 &= \left\langle \frac{1}{n}\sum_{i=1}^n \bar{\psi}(\boldsymbol{y}_i)\otimes\bar{\phi}(\boldsymbol{x}_i), \; \frac{1}{n}\sum_{j=1}^n \bar{\psi}(\boldsymbol{y}_j)\otimes\bar{\phi}(\boldsymbol{x}_j) \right\rangle_{\mathrm{HS}} \\[2mm]
&= \frac{1}{n^2}\sum_{i=1}^n\sum_{j=1}^n \left\langle \bar{\psi}(\boldsymbol{y}_j), \; (\bar{\psi}(\boldsymbol{y}_i)\otimes\bar{\phi}(\boldsymbol{x}_i))\, \bar{\phi}(\boldsymbol{x}_j)\right\rangle_{\mathcal{G}} \\[2mm]
&= \frac{1}{n^2}\sum_{i=1}^n\sum_{j=1}^n \langle\bar{\phi}(\boldsymbol{x}_i),\bar{\phi}(\boldsymbol{x}_j)\rangle_{\mathcal{F}} \; \langle\bar{\psi}(\boldsymbol{y}_j),\bar{\psi}(\boldsymbol{y}_i)\rangle_{\mathcal{G}} \\[2mm]
&= \frac{1}{n^2}\sum_{i=1}^n\sum_{j=1}^n \bar{k}_{ij}\,\bar{l}_{ji} = \frac{1}{n^2}\operatorname{tr}(\bar{\boldsymbol{K}}\bar{\boldsymbol{L}}) = \frac{1}{n^2}\operatorname{tr}(\boldsymbol{K}\boldsymbol{H}\boldsymbol{L}\boldsymbol{H})
\end{aligned}
\tag{37}
$$

In the next section we show that, in the empirical setting, one can work with a *finite-dimensional* feature map obtained from a factorization of the kernel matrix. This representation makes subsequent covariance expressions far more transparent.

## A.4 Finite-Dimensional Feature Map

Let $\mathcal{F}\colon \boldsymbol{x} \mapsto \phi(\boldsymbol{x})$ be an RKHS with reproducing kernel $k(\,\cdot\,,\boldsymbol{x})$. By the Representer Theorem (cf. Eq. (29)), any solution subject to norm constraints admits the form:

$$f = \sum_{i=1}^n \alpha_i\,\phi(\boldsymbol{x}_i) \tag{38}$$

so $f$ lies in the finite subspace $\mathcal{X} := \operatorname{span}\{\phi(\boldsymbol{x}_i)\}_{i=1}^n$. For $f, g \in \mathcal{X}$ we have:

$$\langle f, g\rangle_{\mathcal{F}} = \left\langle \sum_{i=1}^n \alpha_i\phi(\boldsymbol{x}_i), \sum_{j=1}^n \beta_j\phi(\boldsymbol{x}_j) \right\rangle_{\mathcal{F}} = \boldsymbol{\alpha}^\top \boldsymbol{K}\boldsymbol{\beta}, \tag{39}$$

where $\boldsymbol{K}$ is the kernel matrix $K_{ij} = k(\boldsymbol{x}_i, \boldsymbol{x}_j)$. Because $\boldsymbol{K}$ is symmetric positive semi-definite it admits a factorization:

$$\boldsymbol{K} = \boldsymbol{J}\boldsymbol{J}^\top \tag{40}$$

with $\boldsymbol{J} \in \mathbb{R}^{n\times r}$ full rank. Row $i$ of $\boldsymbol{J}$ (denoted by $\boldsymbol{J}_i^\top$) therefore provides an *implicit finite-dimensional feature vector* for $\phi(\boldsymbol{x}_i)$: indeed $\boldsymbol{J}_i^\top \boldsymbol{J}_j = K_{ij} = \langle\phi(\boldsymbol{x}_i),\phi(\boldsymbol{x}_j)\rangle_{\mathcal{F}}$ Moreover:

$$\langle f, g\rangle_{\mathcal{F}} = \boldsymbol{\alpha}^\top \boldsymbol{J}\boldsymbol{J}^\top \boldsymbol{\beta} = (\boldsymbol{J}^\top \boldsymbol{\alpha})^\top(\boldsymbol{J}^\top \boldsymbol{\beta}) = \boldsymbol{u}^\top \boldsymbol{v} \tag{41}$$

where we have set $\boldsymbol{u} = \boldsymbol{J}^\top \boldsymbol{\alpha}$ and $\boldsymbol{v} = \boldsymbol{J}^\top \boldsymbol{\beta}$. Hence $\boldsymbol{u}, \boldsymbol{v} \in \mathbb{R}^r$ are equivalence of $f$ and $g$ in this finite-dimensional feature map. Using the reproducing property, the centred evaluation vector $\bar{\boldsymbol{f}} = [\, f(\boldsymbol{x}_1) - \mu_f, \ldots, f(\boldsymbol{x}_n) - \mu_f\,]^\top$, with $\mu_f = \frac{1}{n}\sum_{i=1}^n f(\boldsymbol{x}_i)$, satisfies :

$$\bar{\boldsymbol{f}} = \boldsymbol{J}\boldsymbol{u} - \boldsymbol{1}\left(\tfrac{1}{n}\boldsymbol{1}^\top \boldsymbol{J}\boldsymbol{u}\right) = \left(\boldsymbol{I} - \tfrac{1}{n}\boldsymbol{1}\boldsymbol{1}^\top\right)\boldsymbol{J}\boldsymbol{u} = \boldsymbol{H}\boldsymbol{J}\boldsymbol{u} \tag{42}$$

where $\boldsymbol{H}$ is the centring matrix. Now let $\mathcal{G}: \boldsymbol{y} \mapsto \psi(\boldsymbol{y})$ with Gram matrix $\boldsymbol{L} = \boldsymbol{D}\boldsymbol{D}^\top$. Repeating the same construction yields $\bar{\boldsymbol{g}} = \boldsymbol{H}\boldsymbol{D}\boldsymbol{v}$ for $g(\boldsymbol{y}) = \sum_{j=1}^n \beta_j \psi(\boldsymbol{y}_j)$. Thus the empirical objective becomes:

$$
\begin{aligned}
\sup_{g \in \mathcal{G}} \sup_{f \in \mathcal{F}} \mathbb{E}\big[\bar{f}(\boldsymbol{x})\,\bar{g}(\boldsymbol{y})\big] &\approx \frac{1}{n}\sum_{i=1}^n \big(f(\boldsymbol{x}_i) - \mu_f\big)\big(g(\boldsymbol{y}_i) - \mu_g\big) \\
&= \frac{1}{n}(\boldsymbol{H}\boldsymbol{D}\boldsymbol{v})^\top(\boldsymbol{H}\boldsymbol{J}\boldsymbol{u}) \\
&= \frac{1}{n}\boldsymbol{v}^\top \boldsymbol{D}^\top \boldsymbol{H}\boldsymbol{J}\boldsymbol{u} \;=\; \boldsymbol{v}^\top \widehat{\boldsymbol{C}}_{yx}\,\boldsymbol{u}
\end{aligned}
\tag{43}
$$

where $\widehat{\boldsymbol{C}}_{yx} = \frac{1}{n}\boldsymbol{D}^\top \boldsymbol{H}\boldsymbol{J}$ is the (finite-sample) cross-covariance matrix corresponding to the empirical cross-covariance operator $\widehat{\mathbb{C}\mathrm{ov}}_{yx}$. In the following section we derive the Equation (11).

## A.5 RKHS Encoders for Feature-Space Alignment

Recall from Section 4.1 the following random $(Z^i = \varepsilon^i(\boldsymbol{\theta}^i; X^i)$:

- $X^i$: encoder input at iteration $i$
- $Z^i$: encoder output at iteration $i$
- $X$: initial representation
- $S$: undesired concept labels
- $Y$: target task labels

Let $\mathcal{F}$ be an RKHS in which we seek encoders $f$ that optimize the following objective:

$$
\begin{aligned}
\sup_{\{g_{\mathcal{I}}\}} \sup_{f} \quad & \mathbb{E}[\bar{g}_{x^i}(X^i)\bar{f}(Z^i)]^2 + \tau_x\,\mathbb{E}[\bar{g}_x(X)\bar{f}(Z^i)]^2 + \tau_y\,\mathbb{E}[\bar{g}_y(Y)\bar{f}(Z^i)]^2 \\
\text{s.t.} \quad & \sup_{g_s} \mathbb{E}[\bar{g}_s(S)\bar{f}(Z^i)] = 0, \quad \|g_{\mathcal{I}}\|_{\mathcal{G}_{\mathcal{I}}} = \|f\|_{\mathcal{F}} = \|g_s\|_{\mathcal{G}_s} = 1
\end{aligned}
\tag{44}
$$

Here, $\mathcal{G}_{\mathcal{I}}$ (with $\mathcal{I} = \{x^i, x, y\}$) and $\mathcal{G}_s$ refer to corresponding RKHSs for $g_{\mathcal{I}}$ and $g_s$. Let the kernel matrices for $Z^i$, $X^i$, $X$, $Y$, and $S$ be factorized as

$$\boldsymbol{K}_{z^i} = \boldsymbol{L}_{z^i}\boldsymbol{L}_{z^i}^\top, \quad \boldsymbol{K}_{x^i} = \boldsymbol{J}_{x^i}\boldsymbol{J}_{x^i}^\top, \quad \boldsymbol{K}_x = \boldsymbol{J}_x\boldsymbol{J}_x^\top, \quad \boldsymbol{K}_y = \boldsymbol{L}_y\boldsymbol{L}_y^\top, \quad \boldsymbol{K}_s = \boldsymbol{L}_s\boldsymbol{L}_s^\top$$

Using the finite-dimensional feature representation discussed earlier, the empirical estimate for each term in the objective, for instance $\mathbb{E}[\bar{g}_y(Y)\bar{f}(Z^i)]$, can be written as:

$$\mathbb{E}[\bar{g}_y(Y)\bar{f}(Z^i)] \approx \frac{1}{n}\sum_{j=1}^n \bar{g}_y(\boldsymbol{y}_j)\bar{f}(\boldsymbol{z}_j^i) = \frac{1}{n}\boldsymbol{u}_y^\top \boldsymbol{L}_y^\top \boldsymbol{H}\boldsymbol{L}_{z^i}\boldsymbol{w} = \boldsymbol{u}_y^\top \widehat{\boldsymbol{C}}_{yz^i}\boldsymbol{w} \tag{45}$$

where $\boldsymbol{u}_y$ and $\boldsymbol{w}$ are the equivalent vectors to $g_y$ and $f$ in the corresponding finite dimensional feature map, and $\widehat{\boldsymbol{C}}_{yz^i}$ denotes the empirical cross-covariance matrix. Note that the following two optimization problems over $\boldsymbol{w}$ are equivalent:

$$\sup_{\boldsymbol{u}_y} \sup_{\boldsymbol{w}} \; (\boldsymbol{u}_y^\top \widehat{\boldsymbol{C}}_{yz^i}\boldsymbol{w})^2 \quad \equiv \quad \sup_{\boldsymbol{w}} \; \big\|\widehat{\boldsymbol{C}}_{yz^i}\boldsymbol{w}\big\|_2^2 = \boldsymbol{w}^\top \widehat{\boldsymbol{C}}_{yz^i}^\top \widehat{\boldsymbol{C}}_{yz^i}\boldsymbol{w} \tag{46}$$

This equivalence follows since the optimal $\boldsymbol{u}_y$ is aligned with $\widehat{\boldsymbol{C}}_{yz^i}\boldsymbol{w}$ and satisfies:

$$\boldsymbol{u}_y = \frac{\widehat{\boldsymbol{C}}_{yz^i}\boldsymbol{w}}{\|\widehat{\boldsymbol{C}}_{yz^i}\boldsymbol{w}\|_2} \tag{47}$$

Hence, the full empirical objective becomes:

$$\sup_{\boldsymbol{w}} \quad \boldsymbol{w}^\top \big( \widehat{\boldsymbol{C}}_{x^i z^i}^\top \widehat{\boldsymbol{C}}_{x^i z^i} + \tau_x \, \widehat{\boldsymbol{C}}_{xz^i}^\top \widehat{\boldsymbol{C}}_{xz^i} + \tau_y \, \widehat{\boldsymbol{C}}_{yz^i}^\top \widehat{\boldsymbol{C}}_{yz^i} \big) \boldsymbol{w} \tag{48}$$

Now consider the constraint in Eq. (44), which ensures that $f$ does not increase alignment with the undesired attribute $S$. Its empirical estimate becomes:

$$\sup_{g_s} \mathbb{E}[\bar{g}_s(S)\bar{f}(Z^i)] \approx \sup_{\boldsymbol{u}_s} \boldsymbol{u}_s^\top \widehat{\boldsymbol{C}}_{sz^i} \boldsymbol{w} = 0 \tag{49}$$

To satisfy this constraint, we require

$$\boldsymbol{w} \in \mathrm{Null}(\widehat{\boldsymbol{C}}_{sz^i})$$

Let $\boldsymbol{Q}$ be an orthonormal basis for this null space. Then we can write:

$$\boldsymbol{w} = \boldsymbol{Q}\boldsymbol{v} \tag{50}$$

Since orthonormal transformations preserve norm, the constraint remains unchanged. Substituting into the objective yields:

$$\sup_{\boldsymbol{v}} \quad \boldsymbol{v}^\top \boldsymbol{Q}^\top \left( \widehat{\boldsymbol{C}}_{x^i z^i}^\top \widehat{\boldsymbol{C}}_{x^i z^i} + \tau_x \, \widehat{\boldsymbol{C}}_{xz^i}^\top \widehat{\boldsymbol{C}}_{xz^i} + \tau_y \, \widehat{\boldsymbol{C}}_{yz^i}^\top \widehat{\boldsymbol{C}}_{yz^i} \right) \boldsymbol{Q}\boldsymbol{v}$$
$$\text{s.t.} \quad \|\boldsymbol{v}\|_2 = 1 \tag{51}$$

This is a Rayleigh quotient maximization problem. The optimal solution $\boldsymbol{v}$ corresponds to the eigenvector of the matrix with the largest eigenvalue. Define:

$$\boldsymbol{A} = \boldsymbol{Q}^\top \left( \widehat{\boldsymbol{C}}_{x^i z^i}^\top \widehat{\boldsymbol{C}}_{x^i z^i} + \tau_x \, \widehat{\boldsymbol{C}}_{xz^i}^\top \widehat{\boldsymbol{C}}_{xz^i} + \tau_y \, \widehat{\boldsymbol{C}}_{yz^i}^\top \widehat{\boldsymbol{C}}_{yz^i} \right) \boldsymbol{Q} \tag{52}$$

The matrix $\boldsymbol{A}$ is a sum of symmetric positive semi-definite matrices, and is itself symmetric and PSD. Using its eigen-decomposition:

$$\boldsymbol{A} = \boldsymbol{D}\boldsymbol{\Lambda}\boldsymbol{D}^\top$$
$$\boldsymbol{v}^\top \boldsymbol{A} \boldsymbol{v} = \boldsymbol{v}^\top \boldsymbol{D}\boldsymbol{\Lambda}\boldsymbol{D}^\top \boldsymbol{v} = \boldsymbol{a}^\top \boldsymbol{\Lambda} \boldsymbol{a} = \sum a_i^2 \lambda_i \leq \lambda_{\max} \tag{53}$$

where $\boldsymbol{a} = \boldsymbol{D}^\top \boldsymbol{v}$ and $\|\boldsymbol{a}\|_2^2 = 1$, since the eigenvectors of symmetric matrices form an orthonormal basis. Subsequent encoders, constrained to be orthogonal to the previously selected ones, can be obtained by iteratively maximizing the same Rayleigh quotient over orthogonal complements. By the *Courant–Fischer* min–max principle, these correspond to the eigenvectors associated with the next largest eigenvalues of $\boldsymbol{A}$. One may then select the top $d$ eigenvectors—ranked by eigenvalue magnitude or a normalized criterion—as encoder directions for the next iteration.

## B   Implementation Details

### B.1   Obliviator's Training Procedure

The practical implementation of Obliviator is outlined in Algorithm 1. Depending on the availability of target task labels, the erasure setting is configured as either supervised (when target labels are available) or unsupervised. In the supervised case, an additional term involving $\tau_y \boldsymbol{K}_y$ is included in the encoder loss (8), and correspondingly in the RKHS-based eigenvalue problem (11) as $\tau_y \widehat{\boldsymbol{C}}_{yz^i}^\top \widehat{\boldsymbol{C}}_{yz^i}$. This term is omitted under the unsupervised setting.

The encoder is then trained for a fixed number of epochs, which must be sufficient to enable effective transfer of information from the original representation. If the encoder is trained for too few epochs, its capacity to preserve relevant features may be limited. In such cases, one practical strategy is to pre-train the encoder for a few epochs without applying the undesired-concept removal term, and then perform erasure for a smaller number of epochs. However, excessive pretraining can also be detrimental: the encoder may learn overly complex representations that are no longer smooth enough to be effectively constrained by the smooth witness functions. This, in turn, may necessitate stronger (and potentially less smooth) witnesses, which can hinder erasure quality. Nonetheless, since initial representations from language models are typically expressive, even random projections can retain

---

**Algorithm 1** Obliviator Training Procedure

---

1: **Input:** data $\{x_i\}$, unwanted labels $\{s_i\}$, optional target labels $\{y_i\}$
2: **for** $j = 1$ to $M$ **do**
3:     **if** $\{y_i\}$ is available **then**
4:         RVs $\leftarrow (x^j, x, y)$                                                             ▷ Supervised
5:     **else**
6:         RVs $\leftarrow (x^j, x)$                                                                ▷ Unsupervised
7:     **end if**
8:     Train encoder $\varepsilon^j$ using loss (8) $\leftarrow$ RVs                             ▷ Imposing Independence
9:     $z^j \leftarrow \varepsilon^j(x^j)$
10:     Solve EVP (11), obtain encoder                                  ▷ RKHS Disentanglement
11:     $x^{j+1} \leftarrow$ encode $z^j$ via (12)
12: **end for**

---

most task-relevant information. [4]Thus, the encoder can often preserve key features with relatively few training iterations. In our experiments, we found that 10–15 full-batch iterations were typically sufficient.

After training the encoder, its output is passed to the eigenvalue problem defined in (11). Since the feature space induced by the kernel can be high-dimensional, even for a moderate number of training samples, explicit factorization of the kernel matrix is generally intractable. To address this, we employ approximation methods such as the Nyström method or Random Fourier Features (RFF) [24] to obtain a finite-dimensional approximation of the feature map. Next, the resulting eigenvalues are normalized by the largest eigenvalue, and the top $m$ eigenvectors are selected based on a predefined threshold. These eigenvectors correspond to functions in the RKHS, which serve as new encoder. We apply these functions to the encoder's output to generate a transformed representation. This transformed representation then becomes the input to the encoder in the subsequent iteration. The implementation of Obliviator can be found at the `Obliviator page`.

## B.2   Datasets

We evaluate on three benchmark datasets commonly used for concept erasure: BIAS IN BIOS, DIAL-SENTIMENT and DIAL-MENTION.

- **BIAS IN BIOS [8]:** This dataset consists of biographical texts, each annotated with a profession (the primary task) and a gender label (the sensitive/protected attribute). It includes 28 distinct professions and two gender categories, with 53.7% of the data associated with male subjects and 46.3% with female subjects. The most common profession in the dataset is "professor," accounting for 30% of the total samples. To ensure a fair comparison with FaRM, we used the same dataset split as FaRM [6] for the fine-tuned BERT representations. For the frozen representations, we followed the dataset split used by [26].

- **DIAL [4]:** This dataset consists of two subsets: **DIAL-SENTIMENT** and **DIAL-MENTION**.

  - **DIAL-SENTIMENT** is labeled for sentiment analysis, with sentiment as the primary target variable (*happy* 54.57%, *sad* 45.43%). It also includes *race* labels (*African-American English* 36.93%, *Standard American English* 63.07%).

  - **DIAL-MENTION** is a binary classification dataset for detecting whether a tweet mentions another user (50% conversational, 50% non-conversational). The *race* labels in this subset are equally distributed (50%-50%).

## B.3   Experimental Setup

We use a multilayer perceptron (MLP) as our encoder, consisting of a single hidden layer with 256 units and the SiLU activation function. Optimization is performed using the AdamW optimizer with default hyperparameters. We set the learning rate to $5 \times 10^{-4}$ and apply a weight decay of 0.001. For the BIAS IN BIOS dataset, the encoder is trained for 30 iterations in the first step and 25 iterations in subsequent steps. Due to the infeasibility of computing the full kernel matrix on the entire dataset,

---

[4]This is consistent with the Johnson–Lindenstrauss lemma.

Table 2: Hyperparameters used for the training of Obliviator across different datasets.

| Dataset | Erasure | Encoder Parameters Eq.(8) | | | EVP Parameters Eq.(11) | | |
| | | $\tau_{x^i}$ | $\tau_x$ | $\tau_y$ | $\tau_x$ | $\tau_y$ | Threshold |
|---|---|---|---|---|---|---|---|
| **BIAS IN BIOS** | Supervised | 0.05 | 0.02 | 4 | 0.2 | 3 | $10^{-4}$ |
| | Unsupervised | 0.1 | 0.05 | – | 0.5 | – | $10^{-4}$ |
| **DIAL-SENTIMENT** | Supervised | 0.05 | 0.02 | 4 | 0.2 | 3 | $5 \times 10^{-5}$ |
| | Unsupervised | 0.1 | 0.05 | – | 0.2 | – | $5 \times 10^{-5}$ |
| **DIAL-MENTION** | Supervised | 0.05 | 0.02 | 4 | 0.2 | 3 | $5 \times 10^{-5}$ |
| | Unsupervised | 0.1 | 0.05 | – | 0.2 | – | $5 \times 10^{-5}$ |

exact kernel-based training is only possible via stochastic gradient descent with a reasonable batch size. However, we observe that with RFF and training on the full-batch consistently yields better performance. Consequently, we approximate the kernel using RFF throughout our experiments. The RFF dimensionality is set to 2500/6000 in the first iteration and reduced to 1500 in later steps, as the encoder's input dimension is reduced after the first transformation. Sigma was chosen based on the median heuristic. To ensure that HSIC is not artificially increased through isotropic scaling, we evaluated two normalization strategies: (1) feature-wise normalization to unit variance and (2) sample-wise normalization. The latter proves more effective in our setup. For the DIAL-MENTION and DIAL-SENTIMENT datasets all settings are the same except the initial training is set to 30 iterations. All reported trade-off curves are over three independent runs where we reported minimum performance (lowest trade-off curve). For the RKHS encoder, the Kernel matrix is approximated using RFF with a dimension of 1500. The hyper-parameters used for training can be found in Table 2. Training is conducted on a single NVIDIA RTX A6000 GPU. For the training of DeepSeek and LLaMa on BIAS IN BIOS dataset all the parameters are similar to Table 2 except we set the threshold to $10^{-5}$.

## B.4 Computational Complexity and Scalability of Obliviator

The complexity of Obliviator is determined by its two steps per iteration:

### Encoder

The primary cost is calculating the empirical HSIC loss. There are two general approaches for this:

- *Direct Kernel Matrix Calculation*: This method's time and memory complexity of $O(N^2)$ (where $N$ is the batch size) becomes intractable for large batches. This forces a trade-off, restricting the method to small batch sizes which can lead to a less accurate empirical estimation of HSIC.

- *RFF Approximation*: To enable large-batch training for a better HSIC estimation, we use a scalable alternative based on Random Fourier Features (RFF). This approach has a time complexity of $O(Nd_{\text{rff}}^2)$ and a more favorable memory complexity of $O(Nd_{\text{rff}})$. The core computations involve large matrix multiplications, which are highly optimized for modern GPU architectures, making this method very efficient in practice.

### Eigen Value Problem (EVP)

This step involves solving an EVP. We again use RFF to make this step tractable. The complexity breaks down as follows:

- Constructing the matrix for the EVP involves matrix multiplications with a complexity of $O(Nd_{\text{rff}}^2)$.

- The eigenvalue solver itself has a complexity of $O(d_{\text{rff}}^3)$.

In typical deep learning setup data size $N$ is larger than the RFF dimension $d_{\text{rff}}$ and the overall cost of this step is likely dominated by the matrix multiplications, making its complexity effectively $O(Nd_{\text{rff}}^2)$.

Table 3: Comparison of theoretical and practical time complexity in each step (BIAS IN BIOS)

| Time Complexity | Encoder | EVP |
|---|---|---|
| **Theoretical** | $\mathcal{O}(N d_{\text{embd}} d_{\text{hidden}} + N d_{\text{rff}} d_{\text{hidden}} + N d_{\text{rff}}^2)$ | $\mathcal{O}(d_{\text{rff}}^3 + N d_{\text{rff}}^2)$ |
| **Average time per step (BERT)** | $177.2\,\text{ms} \pm 0.38\,\text{ms}$ | $172\,\text{ms} \pm 1.23\,\text{ms}$ |
| **Average time per step (DeepSeek)** | $219.6\,\text{ms} \pm 1.4680\,\text{ms}$ | $165\,\text{ms} \pm 1.65\,\text{ms}$ |

Table 4: Comparison of theoretical and practical memory complexity in each step (BIAS IN BIOS)

| Memory Complexity | Encoder | EVP |
|---|---|---|
| **Theoretical** | $\mathcal{O}(d_{\text{embd}} d_{\text{hidden}} + d_{\text{rff}} d_{\text{hidden}} + N d_{\text{rff}} + d_{\text{rff}}^2)$ | $\mathcal{O}(d_{\text{rff}}^2 + N d_{\text{rff}})$ |
| **Utilized Memory** | 8.18 GB | 6.78 GB |

The following tables show the theoretical time/memory complexity of our method, alongside its practical complexity measured on an NVIDIA RTX-A6000 GPU. We also report the total time taken to perform erasure across different datasets, language models, and erasure schemes.

Table 5: Erasure time for Obliviator BIAS IN BIOS-frozen

| BIAS IN BIOS Frozen | 99.4%→70% (sec) | 70%→65% (sec) | 65%→60% (sec) | 60%→RC (sec) | 99.4%→RC (sec) |
|---|---|---|---|---|---|
| **BERT** | | | | | |
| Unsupervised | 21.9±0.05 | 27.4±0.06 | 107.76±0.26 | 323.7±0.7 | 482.9±1.1 |
| Supervised | 27.1±0.06 | 72.26±0.1 | 153.5±0.3 | 487.7±0.5 | 749.7±1.7 |
| **DeepSeek** | | | | | |
| Unsupervised | 74.3±0.5 | 67.53±0.46 | 94.54±0.64 | 189.08±1.28 | 425.4±2.9 |
| Supervised | 156±1 | 100.3±0.67 | 390.1±2.63 | 1070±7.2 | 1716.3±11.6 |

Table 6: Erasure time Obliviator on BIAS IN BIOS-Finetuned

| BERT | 99.4%→70% (sec) | 70%→65% (sec) | 65%→63% (sec) | 63%→RC (sec) | 99.4%→RC (sec) |
|---|---|---|---|---|---|
| Unsupervised | 37.7±0.08 | 62.8±0.14 | 540.7±1.2 | 4225.6±9.35 | 4363.9±9.6 |
| Supervised | 43.2±0.1 | 162.06±0.36 | 108.04±0.24 | 5596±12.4 | 5900±12.2 |

Table 7: Erasure time for Obliviator on DIAL-MENTION-Frozen

| BERT | 80%→70% (sec) | 70%→60% (sec) | 60%→RC (sec) | 80%→RC (sec) |
|---|---|---|---|---|
| Unsupervised | 55.2±0.1 | 193.3±0.5 | 1293.2±3.0 | 1541.67±3.6 |
| Supervised | 126.4±0.3 | 144.5±0.3 | 1372.9±3.7 | 1643.8±3.8 |

Table 8: Erasure time for FARM and KRAM on BIAS IN BIOS

| Method | Batch Size | Memory (MB) | Time (sec/epoch) | Total Time (sec) |
|---|---|---|---|---|
| **FaRM** | 1024 | 96.4 | 21.4±1.2 | 535±30 |
| | 4096 | 305.44 | 13.22±1.03 | 330.5±25.75 |
| **KRaM** | 1024 | 76.8 | 65.58±1.4 | 3279±65.2 |
| | 4096 | 479.04 | 45.81±8.5 | 2290±42.5 |

# C  Probing Networks

## C.1  Our Choice of Probing Networks

Unlike prior work [1, 6], we do not restrict the nonlinear adversary to the default MLP classifier provided by scikit-learn. Instead, we evaluate all methods using two types of adversarial classifiers: an SVM with an RBF kernel (implemented via cuML [25]) and an MLP with two hidden layers of 128 neurons (implemented in PyTorch [21]). The MLP classifier is trained three times with different random seeds, while the SVM is evaluated under seven hyperparameter settings $(\gamma, C) \in \{(10, 5), (10, 10), (5, 10), (5, 5), (1, 5), (1, 1), (0.5, 1)\}$. We report the maximum accuracy obtained across all ten runs (three MLP + seven SVM) for each method. For the target task classifier, we employ the same architecture as the adversarial MLP, a two-layer Multi-Layer Perceptron with 128 neurons per hidden layer. An SVM with a large kernel scale ($\gamma = 10$) and hard margin ($C = 10$) serves as a coarse decision boundary, useful for testing whether erasure occurs by overlapping distributions across unwanted attributes. We must note that, the erasure process is carried by a deterministic function and only shuffling data samples across unwanted attributes is not sufficient for erasure as this shuffling can be still invertible. Therefore, true erasure happens upon distribution matching.

In our experiments, we observed that applying dimensionality reduction often improved adversarial accuracy. For instance, using a strong encoder—an MLP with four hidden layers(see §D.1)—without any iterative refinement led our MLP adversaries to perform at random chance level while SVM with $\gamma = 10, C = 10$ reported around 80% accuracy. However, when we applied RKHS refinement via (11) as a dimensionality reduction step and re-evaluated the MLP adversaries again, their accuracy was similar to that of SVM with $\gamma = 10, C = 10$. This indicates that sensitive information was still present in the representation, and the prior adversary failures were primarily due to training difficulty rather than successful erasure.

## C.2  Ablation Studies with Different Probing Networks

To evaluate the robustness of our probing setup, we assess the trade-off curves using multiple types of classifiers. Specifically, we consider a Multi-Layer Perceptron (MLP) with five hidden layers of size 128, as well as Random Forest classifiers with 100 estimators and maximum depths of 15, 20, and 25. For the MLP, we report the highest accuracy across three independent runs. For the Random Forest, we report the best accuracy obtained across the three depth settings. The resulting trade-off curves are shown in Figure 8a, alongside the curve obtained using our default probing classifier (as described earlier). As illustrated in the figure, our choice of classifier yields a consistently lower trade-off profile, demonstrating its robustness for evaluating erasure performance.

In contrast to prior work, we do not rely on visualizations produced by t-SNE or UMAP to assess overlap with respect to the sensitive attribute (e.g., gender). While such techniques can be visually appealing, they are unreliable indicators of structure in high-dimensional spaces, often distorting distances and cluster separability. Instead, we adopt an evaluation based on the training accuracy of an expressive classifier. In particular, if a sufficiently flexible probing model is unable to overfit the training data, this provides stronger evidence that the representation has been overlapped across the unwanted attribute, suggesting successful erasure. However, it is important to note that we are working with finitely many samples in a high dimensional space. It is unlikely that the data points become so close that no probing classifier can overfit to training samples. For this reason, the flexibility of the probing model must be chosen reasonably so that this evaluation becomes meaningful. Figure 10 presents the train and test accuracy of the MLP with five hidden layers over the course of erasure. As the process progresses, we observe a drop in both training and test accuracy, with the training accuracy falling to near random chance. This indicates that even a high-capacity classifier cannot overfit to the training samples, suggesting that the representation has been successfully aligned across the sensitive groups via distributional overlap.

Taken together, these probing results indicate that Obliviator consistently achieves full concept erasure. This is consistent with the theoretical motivation: minimizing $\text{HSIC}(S, X)$ enforces statistical independence between the representation $X$ and the sensitive attribute $S$.

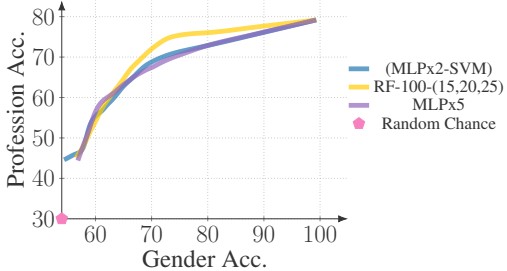

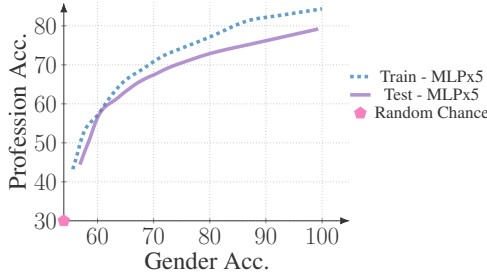

(a) Comparison of obtained trade-off with deeper MLP (5 hidden layers) and Random Forest with our choice of classifier (MLPx2-SVM).

(b) Train and test accuracy for MLP with 5 hidden layer at different step of the erasure.

Figure 8: Ablation studies with different probing networks. Dataset is BIAS IN BIOS and language model is BERT.

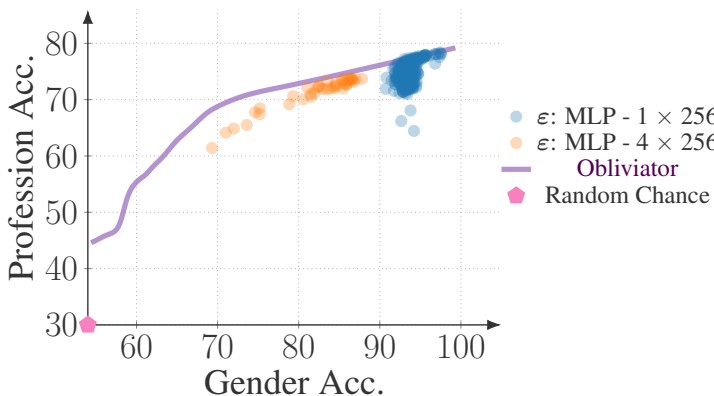

Figure 9: Multi-Step Erasure vs. Single-Step Erasure. Comparison of erasure performance using a single-step encoder with either 1 or 4 hidden layers (MLP), versus the proposed multi-step erasure framework (Obliviator). Results are shown on the BIAS IN BIOS dataset using BERT representations. The multi-step approach consistently demonstrates more effective erasure.

# D    Additional Results

## D.1    Single Step Erasure Vs Multi-Step Erasure

To highlight the importance of each component in Obliviator, we evaluate a simplified baseline using a single MLP encoder with either one or four hidden layers (each containing 256 neurons). The goal is to perform erasure using only the encoder trained with the loss in (8), without employing the multi-step framework or RKHS-based refinement. This setup allows us to isolate the contribution of these components and assess their effect on erasure and adversarial optimization. During training, we evaluate both the target task and undesired attribute accuracy every 30 steps. As shown in Figure 9, neither of the single-step encoder configurations achieves full concept erasure. Although both gender and profession accuracies initially decline, they eventually plateau and begin to oscillate, failing to reach full erasure. While a more expressive encoder might hypothetically improve performance, a direct comparison reveals that the empirical trade-off achieved by Obliviator is consistently better. This demonstrates that the multi-step procedure and RKHS refinement, results in a more robust and effective erasure process.

## D.2    Further Results on Fairness

Here we show the additional fairness for controlled setup experiments 5.3. Moreover, we show the fairness for BIAS IN BIOS and DIAL-MENTION before and after erasure.

Table 9: Erasure with Obliviator on DeepMoji representations in controlled setup. Split describe levels of unwanted attribute disproportion.

| DeepMoji Rep. | Race Acc. | Split | | | |
|---|---|---|---|---|---|
| | | **50%** | **60%** | **70%** | **80%** |
| **Sentiment Acc($\uparrow$)** | Original | 75.38 | 75.57 | 74.16 | 72.37 |
| | 75% | 75.28 | 74.68 | 73.89 | 70.44 |
| | 65% | 74.83 | 72.94 | 68.75 | 64.06 |
| | 55% | 74.53 | 69.62 | 62.82 | 57.82 |
| **DP($\downarrow$)** | Original | 0.1078 | 0.2266 | 0.3109 | 0.3751 |
| | 75% | 0.0667 | 0.1028 | 0.2896 | 0.2293 |
| | 65% | 0.0280 | 0.0330 | 0.0848 | 0.0203 |
| | 55% | 0.0237 | 0.0600 | 0.0650 | 0.0671 |
| $\text{Gap}_{\text{RMS}}(\downarrow)$ | Original | 0.1360 | 0.2412 | 0.3212 | 0.3815 |
| | 75% | 0.1026 | 0.1324 | 0.2991 | 0.2350 |
| | 65% | 0.0920 | 0.0951 | 0.1251 | 0.0917 |
| | 55% | 0.0910 | 0.1050 | 0.0921 | 0.0915 |

Table 10: DP and $\text{Gap}_{\text{rms}}$ for BIAS IN BIOS Frozen Representations

| | Method | BERT | DeepSeek |
|---|---|---|---|
| **DP($\downarrow$)** | Original | 0.0204 | 0.0205 |
| | Obliviator | 0.0147 | 0.0120 |
| $\text{Gap}_{\text{rms}}(\downarrow)$ | Original | 0.1573 | 0.1522 |
| | Obliviator | 0.0464 | 0.0662 |

Table 11: DP and $\text{Gap}_{\text{rms}}$ for DIAL-MENTION Frozen Representations

| Method | **DP($\downarrow$)** | $\textbf{Gap}_{\textbf{rms}}(\downarrow)$ |
|---|---|---|
| Original | 0.219 | 0.2109 |
| Obliviator | 0.006 | 0.0419 |

### D.3 Hyperparameter Sensitivity

We conduct two additional studies to evaluate the impact of hyperparameters on the performance of Obliviator. First, we examine the effect of the RBF kernel bandwidth parameters $\gamma_{z^i}$ and $\gamma_{x^i}$, which are defined as the inverse of the kernel width (i.e., $\gamma = 1/2\sigma^2$). These parameters control the smoothness and expressivity of the witness functions used in dependency estimation. Intuitively, higher $\gamma$ values (corresponding to narrower kernels) yield more complex witness functions, which are more sensitive to fine-grained correlations. This can result in more aggressive erasure by capturing spurious dependencies. This behavior is consistent with the patterns observed in Figure 10a. Furthermore, we can see that Obliviator performs consistently across a broad, reasonable range of values for $\gamma$, demonstrating robustness to this choice. Note that, in this experiment, we considered $\gamma = \gamma_{z^i} = \gamma_{x^i}$.

Next, we study the effect of the weighting coefficients $\tau_{x^i}$ and $\tau_x$, which appear in both the adversarial loss (8) and the eigenvalue problem in (11). In this experiment, we vary only the encoder loss parameters while keeping the EVP parameters fixed, as specified in Table 2. As before, we set $\tau_{x^i} = \tau_x$. The results, shown in Figure 10b, indicate that very large values of $\tau$ lead to a slight degradation in performance. Moreover, increasing $\tau$ slows convergence, requiring more iterations to reach the same level of erasure.

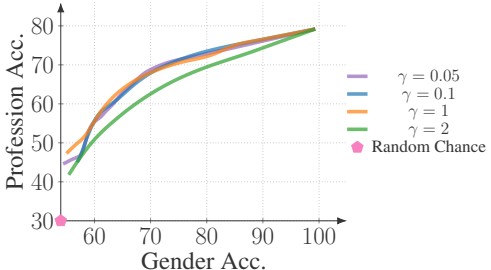
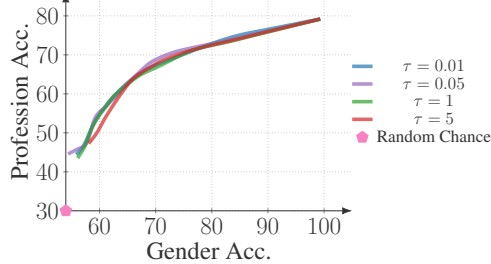

(a) Trade-off obtained by different values of $\gamma = \gamma_{xi} = \gamma_{zi}$.

(b) Trade-off obtained by different values of $\tau = \tau_{xi} = \tau_{zi}$. Here we only change the encoder's parameter and $\tau$ in EVP problem are same as in Table 2.

Figure 10: Analysis of the effect of hyperparameters on Obliviator's performance. Dataset is BIAS IN BIOS and language model is BERT.

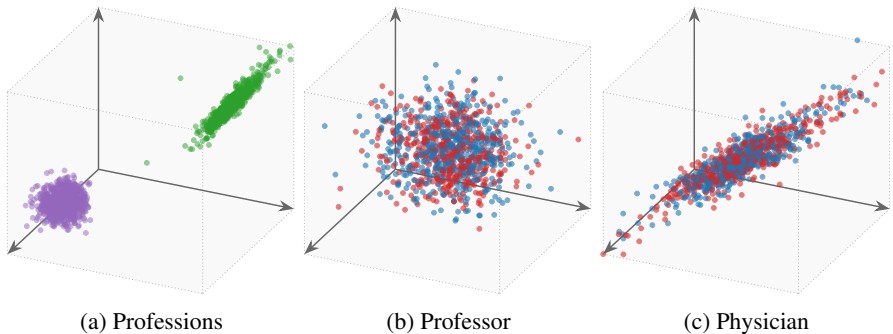

(a) Professions         (b) Professor         (c) Physician

Figure 11: Representations Learned by Obliviator on Finetuned BIAS IN BIOS Representations (BERT). The professions Professor and Physician are shown separately to better visualize the distribution of gender within each class. While the two professions are clearly separated (green and purple), gender labels (blue and red) are indistinguishable within each profession—indicating that Obliviator effectively erases gender information while preserving task-relevant structure.

## D.4 Visualization

Here, we visualize the representations of two professions—Professor and Physician—from the BIAS IN BIOS dataset using fine-tuned representations at an intermediate stage of erasure. This representation corresponds to a point on the trade-off curve where gender classification accuracy is approximately 60%, while profession classification accuracy remains around 85% (see Figure 4c). The visualization in Figure 11 also corresponds to this stage of erasure.

At this point, the number of RKHS encoders obtained from (11) is three, meaning we visualize the actual transformed representation used by the model. Notably, while the two professions remain well-separated, the gender labels within each profession are indistinguishable—highlighting that Obliviator successfully removes gender information while preserving task-relevant structure.

## E   Limitations

Obliviator requires precise definitions of unwanted attributes to effectively erase associated concepts. However, such definitions may be unavailable or vary across cultural and ethical contexts. Pseudo-labeling using existing AI models can partially mitigate this, but such labels may be noisy and impact model performance, a factor not investigated in this work. Similar limitations apply to supervised erasure, which assumes access to target task labels.

Our method affords a more stable optimization by leveraging tractable components: HSIC to quantify statistical dependency and RKHS disentanglement to make task-relevant information more accessible during optimization. Our experiments confirmed the more utility-preserving erasure of Obliviator

compared to baselines across various setups. In some scenarios, it achieved full erasure without sacrificing utility, setting a strong benchmark for the erasure-utility trade-off. Nevertheless, our method solves the non-convex optimization in (4.1) numerically, and a closed-form, theoretical guarantee for optimal nonlinear erasure remains an open problem.

# F    Societal Impact

Obliviator enables targeted concept erasure from learned representations, which can be beneficial—for instance, by removing sensitive demographic attributes to reduce reliance on them in decision-making, or by erasing personal identifiers to protect privacy. However, if task-critical attributes are treated as "unwanted", such as age in healthcare or income level in finance, model performance may degrade, or the model may rely on spurious correlations with remaining features like gender or race. Transparent documentation, domain-informed definitions, and independent oversight are essential, especially in high-stakes applications. Without careful review, erasure may discard socially meaningful information or fail to generalize across contexts.

