# OpenReview forum: "Obliviator Reveals the Cost of Nonlinear Guardedness in Concept Erasure"
_NeurIPS.cc/2025/Conference — NeurIPS 2025 poster_

### Official Review · Reviewer_eWgU · 2025-06-29

**Clarity:** 3
**Significance:** 3
**Originality:** 3
**Rating:** 5
**Confidence:** 4

**Summary:**

The paper presents Obliviator, a method that tackles the long-standing problem that "concept-erasure" methods often hide sensitive signals (like gender or race) only from linear probes while leaving them exposed to more powerful nonlinear classifiers. Building on a objective that simultaneously minimizes the Hilbert–Schmidt independence between the learned representation and the sensitive attribute while preserving task-relevant information, Obliviator iteratively manipulates the feature space to remove the information, and enables to trace a full erasure-utility curve along the iterations. Across BiasInBios and two dialect-sentiment benchmarks, it drives sensitive-attribute accuracy down to chance while matching or beating task performance of prior linear and nonlinear baselines, thereby offering a post-hoc nonlinear erasure method.

**Questions:**

1. BiasIbBios shows a different utility-erasure curve (in fig 4) than the two other datasets. Why is that? are profession and gender more inhernetly related to each other than dialect and mention/sentiment?

2. Related to the previous question, have you considered running a controlled experiment where you modify the proportion of each sensitive group within in main-task class (or vice versa)?

3. Have you considered running ablations on both the two steps in iterative methods, and also on the different loss components? are they are actually necessary?

4. If I understand correctly the appendix, the paper uses RBF kernel and yet the erasure generalizes well to a strong MLP adversary. This seems to contradict [3] where it was reported that nonlinear erasure does not generalize well between kernels. Do you have thoughts on the differences?

[3] Ravfogel, S., Vargas, F., Goldberg, Y., & Cotterell, R. (2022, December). Adversarial concept erasure in kernel space. In Proceedings of the 2022 Conference on Empirical Methods in Natural Language Processing (pp. 6034-6055).

**Ethical Concerns:**

["NO or VERY MINOR ethics concerns only"]

**Final Justification:**

The problem this paper tackles is important and widely studied, and the approach is creative. The new fairness results in the rebuttal support the empirical relevance of the new method.

**Limitations:**

Yes

**Quality:**

3

**Strengths And Weaknesses:**

The problem this paper tackles is important and widely studied, and the approach is creative, where in step (i) a flexible encoder pushes the representations to regions where the RKHS separates "good" information from "bad" information, and in step (ii) a closed-form operation uses the current features to orthogonalize the sensitive information (as is done in LEACE and other methods). The paper backs this with broad experiments, showing that at every point on the utility-erasure curve Obliviator preserves more or similarly degree of task accuracy than all baselines, and that its ability to exploit task labels in a purely post-hoc setting further enhances that margin.

Concept erasure is usually aiming to achieve on of two goals: (i) making the model's prediction independent or less related to the erased concept, (e.g., gender), thereby achieving a more fair model (e.g., [1]); (ii) using the erasure as an *intervention* technique, that surgically changes the forward pass with the goal of understanding how computation is carried out in the model (e.g., [2]). While the experiments are supporting the main claims of the paper -- achieving nonlinear erasure while maintaining better erasure-accuracy tradeoff than previous linear erasure methods -- they do not test the impact of inducing nonlinear erasure on end-metrics such as fairness of the resulting classifiers (e.g., true positive rate gaps), nor do they use them to interpret the model. Does nonlinear erasure translate to a more "fair" model (under some notion of fairness)? does it allow us to better estimate the causal effect of concepts on the model? I think that including some experiments that would emphasize the utility of the proposed method will improve the paper.

I think the experimental section can be further strengthened by some additional ablation and a more controlled experimental pipeline, see the "questions" section.

[1] Bolukbasi, T., Chang, K. W., Zou, J. Y., Saligrama, V., & Kalai, A. T. (2016). Man is to computer programmer as woman is to homemaker? debiasing word embeddings. Advances in neural information processing systems, 29.

[2] Elazar, Y., Ravfogel, S., Jacovi, A., & Goldberg, Y. (2021). Amnesic probing: Behavioral explanation with amnesic counterfactuals. Transactions of the Association for Computational Linguistics, 9, 160-175.

---

> ### Author Rebuttal · Authors · 2025-07-31
>
> # Summary of the Review & Response
>
> We thank the reviewer for their positive and insightful feedback. We are encouraged that they recognize the *importance* of the problem and find our approach *creative,* backed by *broad experiments (S1)*. The reviewer also highlighted our key strength, *preserve a more or similar degree of task accuracy* than all baselines(S2).
>
> The reviewer's primary concerns are:
>
> 1. Difference between utility-erasure curves across different datasets. (Q1)
> 2. Controlled setup experiment modifying the proportion of sensitive groups (Q2)
> 3. Ablation studies on our two-step solution and its loss components (Q3)
> 4. Contradiction with prior work on generalization to nonlinear adversaries (Q4)
> 5. Reporting fairness metrics (W1)
>
> Our responses are summarized as:
>
> 1. We clarify that the utility-erasure trade-off reflects the *model’s learned dependencies*, not intrinsic properties of the task. We show that newer models trained on BiasInBios exhibit significantly less trade-off than BERT, suggesting BERT had learned strong spurious correlations.
> 2. We performed the proposed experiment on DeepMoji, varying sensitive group proportions. Results confirm that skewed sampling worsens the trade-off due to biased HSIC estimation.
> 3. We conducted ablations on each loss component and the two-step procedure, demonstrating their joint necessity. We also contrast our HSIC-based approach with Kernelized R-LACE, showing that our method defends against a *cascading kernel* adversary and resolves the generalization limitations of prior work.
> 4. We now report *Demographic Parity (DP)* and *GAP*, showing that *Obliviator’s* strong statistical erasure improves downstream fairness.
>
> # Response to Questions:
>
> > ### **Q1) Inherent Trade-off in Utility-Erasure**
> >
>
> The differing trade-off curves across datasets and models reflect the causal challenges we discussed with Reviewer (BjYo-Q1). The trade-off revealed by *Obliviator* serves as a *diagnostic of the model’s learned representation*, including any spurious correlations acquired during pretraining. Since HSIC captures all statistical dependencies, *Obliviator* erases any such entanglements, regardless of their origin.
>
> We hypothesize that the sharper trade-off observed for BiasInBios on BERT (Fig. 4c) arises not from a real-world causal link between gender and profession, but from a *spurious, model-specific correlation*. This is supported by the fact that newer models like *DeepSeek* and *LLaMA* show a milder trade-off on the same dataset (Fig. 6).
>
> As you noted, the trade-off is also much flatter for BERT on the DIAL datasets (Fig. 3), suggesting that in BERT’s representation, *gender and profession* are more entangled than *race and sentiment/mention*.
>
> Ultimately, post-hoc erasure does not measure an objective causal relationship, but how a specific model *internalizes* that relationship, shaped by its training data and inductive biases.
>
> > ### **Q2) Controlled Experiment**
> >
>
> We ran the suggested controlled experiment, hypothesizing that an imbalanced erasure dataset would worsen the utility-erasure trade-off due to a biased HSIC estimate. To test this, we varied the proportion of the Race attribute within Sentiment classes on the DeepMoji and measured the resulting impact on sentiment accuracy at different levels of race accuracy.
>
> ### **TR8- Controlled Setup**
>
> | Race Acc. | 50% | 60% | 70% | 80% |
> | --- | --- | --- | --- | --- |
> | original | 75.38 | 75.57 | 74.16 | 72.37 |
> | 75% | 75.28 | 74.68 | 73.89 | 70.44 |
> | 65% | 74.83 | 72.94 | 68.75 | 64.06 |
> | 55% | 74.53 | 69.62 | 62.82 | 57.82 |
>
> As dataset imbalance increases, the utility-erasure trade-off becomes more severe. *Obliviator* preserves utility well on the balanced split but shows a significant performance drop at 80% imbalance for similar erasure. This finding highlights the inherent challenge of post-hoc erasure: the quality of the erasure and its ability to reflect a true causal graph between concepts is entirely dependent on how well the erasure data represents the real world, and how the model itself has encoded those relationships.
>
> > ### **Q3,Q4) Ablation on Multi-Step Erasure and Differences with Kernelized R-LACE**
> >
>
> ### **Differences with Kernelized R-LACE:**
>
> As outlined in our response to Reviewer (X9EX-Q3) and Reviewer (BjYo-Q2),  the key distinction between *Obliviator* and prior kernelized methods lies in the theoretical guarantees underpinning the approach. *Obliviator* is based on a formal statistical measure of independence (HSIC), while prior work is based on a specific adversarial game.
>
> Kernelized R-LACE is a "single-layer" kernelization. It first maps the representation $X$ into a feature space via $\phi(X)$ (Mercer Feature Map) and applies a linear adversarial game in that space, ensuring that the resulting representation has no linear dependency on the sensitive attribute extracted by a linear adversary in the $\phi(.)$ space. This does not prevent another adversary from applying a different nonlinear map, say $\psi(·)$ and reveal a form of dependency. Thus it uses a Linear test of $(φ(X), S) = 0$  and not a linear test of $(ψ(φ(X)), S) = 0$. From a statistical perspective, the former corresponds to mean independence while the later corresponds to full statistical independence.
>
> Our approach generalizes because it is based on the HSIC principle. Dependence between $X$ and $S$ exists if any functions, $f$ and $g$, can be found that make the relationship between $f(X)$ and $g(S)$ linear. The critical question is: how can we account for *all possible* functions an adversary might use? We need a space of functions that is expressive enough to approximate any potential adversary. An RKHS with a characteristic kernel has this property, as it can approximate any continuous function in an L2 norm. This is precisely how our Eq.(6) formulates the problem: it learns a guarding function $ε$ such that no adversary $f$ from this expressive function space can linearize the relationship between $f(ε(X))$ and the sensitive attribute $S$.
>
> Our method's improved generalization comes from solving a harder problem than Kernelized R-LACE. We address the "cascading kernel" problem, seeking a representation ε(X) that ensures ψ(ε(X)) is not linearly separable from S for any adversarial feature map ψ(·). This is a stricter condition without a closed-form solution, and our method's ability to solve it is why it generalizes.
>
> ### **Ablation on Multi-Step Erasure:**
>
> We performed ablation studies on our method's loss components and two main steps. To test the utility-preserving terms $τ_x$ and $τ_{x^i}$ we set $τ_x=0$ in an experiment on the BiasInBios dataset. The table below shows the impact on task accuracy across different erasure levels:
>
> ### **TR9- Ablation on Loss Parameters**
> | Gender Acc. | 55% | 60% | 70% | 80% | 90% |
> | --- | --- | --- | --- | --- | --- |
> | $τ_x=0.05$ | 45.06 | 55.43 | 68.92 | 73.01 | 76.10 |
> | $\tau_x=0$ | 40.58 | 48.04 | 66.27 | 72.5 | 76.01 |
>
> ### **TR10- Ablation on RKHS Refinement**
> | Gender acc. | 80% | 70% | 65% |
> | --- | --- | --- | --- |
> | RKHS-Refinement | 72.92 | 65.17 | 58.46 |
> | Obliviator | 73.12 | 68.92 | 62.80 |
>
> The utility-preserving term is most critical in the final stages of erasure, where interventions must be more precise to maintain performance.
>
> Our ablation studies show that each of the two steps in our framework is insufficient on its own.
> - Adversarial Training, as shown in Appendix D.1 (Figure 9), results in an unstable optimization that is difficult to tune for complete erasure.
> - RKHS Refinement is less efficient at preserving utility compared to the *Obliviator* framework.
>
> These ablations demonstrate a clear synergistic effect. While the RKHS-refinement step alone works by successively removing modes of dependence that become linear in each new space, it lacks a theoretical guarantee of converging to a full erasure in finitely many steps (required 80 steps to reach 70% accuracy). In contrast, obliviator is backed by HSIC (Equation 6), and two steps makes it highly effective in practice (requiring only 4 iterations for the same result).
>
> # Response to Weaknesses:
>
> > ### **W1 ) Ablation Result on Fairness**
> >
> We hypothesize that enforcing statistical independence between a representation and a sensitive attribute will improve downstream model fairness. Theoretically, as our method drives HSIC → 0, fairness metrics like Demographic Parity **(DP)** should also approach 0. To test this, we evaluated how DP and GAP change at different erasure levels in our controlled DeepMoji experiment.
>
> ### **TR11- DP and $Gap_g^{RMS}$ for Controlled Setup**
>
> | DeepMoji Rep. | Race Acc. | 50% | 60% | 70% | 80% |
> | --- | --- | --- | --- | --- | --- |
> | DP($\downarrow$) | Original | 0.1078 | 0.2266 | 0.3109 | 0.3751 |
> |  | 75% | 0.0667 | 0.1028 | 0.2896 | 0.2293 |
> |  | 65% | 0.028 | 0.033 | 0.0848 | 0.0203 |
> |  | 55% | 0.0237 | 0.06 | 0.0650 | 0.0671 |
> | $Gap_g^{RMS}(\downarrow)$ | Original | 0.1360 | 0.2412 | 0.3212 | 0.3815 |
> |  | 75% | 0.1026 | 0.1324 | 0.2991 | 0.2350 |
> |  | 65% | 0.0920 | 0.0951 | 0.1251 | 0.0917 |
> |  | 55% | 0.0910 | 0.1050 | 0.0921 | 0.0915 |
>
> The fairness metrics for supervised erasure on frozen representation across different dataset and language models reported below.
>
> ### **TR12- DP and  $Gap_g^{RMS}$ for BiasInBios Frozen**
>
> |  | Method | BERT | DeepSeek |
> | --- | --- | --- | --- |
> | DP($\downarrow$) | Original | 0.0204 | 0.0205 |
> |  | Obliviator | 0.0147 | 0.012 |
> | $Gap_g^{RMS}$($\downarrow$) | Original | 0.1573 | 0.1522 |
> |  | Obliviator | 0.0464 | 0.0662 |
>
> ### **TR13- DP and  $Gap_g^{RMS}$ for Dial-Sentiment Frozen**
>
> | Method | DP($\downarrow$) | $Gap_g^{RMS}(\downarrow)$ |
> | --- | --- | --- |
> | Original | 0.219 | 0.2109 |
> | Obliviator | 0.006 | 0.0419 |
>
> ---
> **If our responses addressed your initial comments, please consider raising the score. If you have more questions, we would be happy to answer them.**

---

> > ### Comment · Reviewer_eWgU · 2025-08-03
> > **Response**
> >
> > I appreciate the response that fully answers my concerns. I hope the new results -- and particularly the fairness metrics -- will be incorporated in the final version. I increased my score accordingly.

---

> > > ### Author Response · Authors · 2025-08-04
> > > **Response**
> > >
> > > Thank you for re-evaluating our work and for the positive feedback on our rebuttal. We are very grateful for your engagement.
> > >
> > > We absolutely confirm that all the new results, particularly the downstream fairness metrics you suggested, will be incorporated into the final version of the paper to ensure the work is as complete as possible.
> > >
> > > Thank you again for your valuable guidance throughout this process.

---

### Official Review · Reviewer_BjYo · 2025-06-30

**Clarity:** 3
**Significance:** 3
**Originality:** 3
**Rating:** 5
**Confidence:** 3

**Summary:**

Given the rapid progress of LLMs learning almost everything from the web data, Concept erasure is a key field focused on removing sensitive attribute information from learned representations while preserving their utility for intended tasks. To this end, the authors propose Obliviator, a post-hoc concept erasure method designed to remove attributes like social or demographic factors from learned representations of LLMs using supervised and fine-tuning strategies. It introduces a multi-step framework that progressively transforms the feature space to enable smooth concept removal through adversarial optimization, where its key aspect is its ability to capture nonlinear statistical dependencies between learned representations and undesired attributes, addressing a significant limitation of existing methods.

**Questions:**

Please see the Strengths and Weaknesses section for more details.

**Ethical Concerns:**

["NO or VERY MINOR ethics concerns only"]

**Final Justification:**

The paper introduces a novel post-hoc concept erasure method designed to remove protected attributes from learned representations of LLMs. I vote for accepting this work.

**Limitations:**

Please see the Strengths and Weaknesses section for more details.

**Paper Formatting Concerns:**

Not applicable.

**Quality:**

3

**Strengths And Weaknesses:**

**Strengths**

1. The key strength of the proposed work is that, in contrast to existing methods, it effectively guards against nonlinear adversaries, which have been shown to circumvent existing nonlinear erasure techniques.

2. The proposed method shows strong performance across different families of large language models, maintaining its effectiveness in nonlinear guardedness and performance.

**Weaknesses and Open Questions**

1. For frontier models, we don't know anything about the training data. How does one understand the existence of any confounders for the concepts that need to be erased?

2. Given that Obliviator is a multi-step framework that uses multiple updates, it would be great if the authors could comment on the computational complexity of their proposed algorithm vs. the baselines.

3. The authors raise the question of the "cost of concept erasure," but the cost of erasure analysis is limited to utility performance and is underexplored.

4. How does the proposed algorithm specifically utilize RKHS to capture nonlinear statistical dependencies, and what advantages does this offer over existing linear or less robust nonlinear methods?

5. Minor: It will be beneficial for the reader if the authors could add some implementation description of Obliviator in the main document.

---

> ### Author Rebuttal · Authors · 2025-07-31
>
> # Summary of the Review & Response
>
> We thank the reviewer for their positive and insightful feedback. We are encouraged that they find concept erasure to be a *key field*  (summary) and that they recognize the core strengths of our work: its ability to *capture nonlinear statistical dependencies* (summary) , *effectively guard against nonlinear adversaries (S1)*, and show *strong performance across different families of large language models (S2).*
>
> The reviewer's primary concerns are:
>
> 1. Erasure on frontier models and unobserved confounders (Q1)
> 2. Computational Complexity  (Q2)
> 3. Limiting the cost of erasure to utility performance (Q3)
> 4. Utilization of RKHS for capturing nonlinear dependencies (Q4)
> 5. Minor: Implementation detail (Q5)
>
> Our responses are summarized as follows :
>
> 1. We clarify that the utility-erasure trade-off is a diagnostic of the specific model's learned dependencies, not necessarily an inherent real-world link. We discuss the causal nuances of unobserved confounders versus spurious correlations learned by the model, connecting these concepts to our empirical results comparing BERT and newer models like DeepSeek.
> 2.  We provide a theoretical complexity analysis for each component of *Obliviator*, our RFF-based design ensures obliviator is both scalable and practical. We also refer to our response to Reviewer [4SEV] for detailed empirical benchmarks.
> 3. We explain that our focus on the utility-erasure trade-off addresses the most fundamental axis of cost. We clarify that our work provides the best-known **empirical trade-off**, which is a foundational prerequisite for studying other costs of interest, like perplexity, etc.
> 4.  We detail the twofold role of RKHS in our framework: (1) to robustly **measure** nonlinear dependence via HSIC, and (2) in our novel **refinement step**, to **guide** the optimization by constructing optimal encoders that create a better-conditioned feature space for the next iteration.
> 5.  The implementation details are in Appendix B. To aid reproducibility, we will update Appendix B with a PyTorch code snippet and also make the full implementation code publicly available upon publication.
>
> # Response to Questions:
>
> > ### **Q1) Erasure on Frontier Model and Unobserved Confounders**
> >
>
> A fundamental challenge for any post-hoc erasure method operating on frontier models is the gap between statistical dependence and causal influence.
>
> Our method, *Obliviator*, is designed to enforce statistical independence between the representation $X$ and the sensitive attribute $S$, based on the data provided for erasure. The effectiveness of this process hinges on whether that data accurately reflects the dependencies we aim to remove. This leads to two scenarios, as your comment suggests:
>
> 1. Missing Confounder: If a confounder $C$ influences both $X$ and $S$ (i.e., $C → X$ and $C → S$), but our erasure dataset contains no variation in $C$, our method will be blind to this causal pathway. Enforcing independence on the biased sample may not lead to robust erasure on the data where it is influenced by $C$. This is an inherent limitation of any method that learns from a finite, and potentially unrepresentative, sample of data.
> 2. Spurious Correlations from Model Training: If the frontier model's original training process encoded a spurious correlation between the target label $Y$ and the sensitive attribute $S$ into the representation $X$, our method will also treat the resulting dependency between $X$ and $S$ as something to be removed. This may force the erasure to be more aggressive than is causally necessary, potentially leading to a greater loss of utility.
>
> However, it is important to note that since HSIC is agnostic to causal origin, it will account for all observable dependencies, whether they are direct, spurious, or confounded. Therefore, our erasure is statistically complete if the samples used for erasure are representative of the real-world data distribution.
>
> This distinction may help explain one of our key empirical findings. We hypothesize that the sharper erasure trade-off observed on BERT compared to DeepSeek (Figure 6b) could be because older models like BERT have stronger spurious correlations between representations and attributes. Newer, more capable models like DeepSeek may have representations that are already better disentangled, making the erasure of the remaining statistical dependence less costly to the representation's utility.
>
> > ### **Q2) Computational Complexity**
> >
>
> We have provided a detailed empirical breakdown of the computational cost (including wall-clock times and memory usage) and a comparison to baselines in our response to Reviewer[4SEV] (see tables TR1-TR5), which will be added to the appendix for your reference. For convenience, we summarize the theoretical complexity here.
>
> The primary cost is calculating the empirical HSIC loss. There are two general approaches for this:
>
> - Direct Kernel Matrix Calculation: This method's time and memory complexity of $O(N^2)$ becomes intractable for large datasets. This restricts the method to small datasets, which can lead to a less accurate estimation of HSIC.
> - RFF Approximation: To enable full-batch training for a better HSIC estimation, we use a scalable alternative based on RFF. This approach has a time complexity of $O(N⋅d_{rff}^x.d_{rff}^y + N⋅d_{rff}^x.d_{rff}^s )$ and better memory complexity of $O(N⋅d_{rff})$.
>
> For EVP step we use RFF to make this step tractable. The complexity breaks down is:
>
> - Constructing the matrix for the EVP involves  $O(N⋅d_{rff}^x.d_{rff}^y + N⋅d_{rff}^x.d_{rff}^s )$.
> - The eigenvalue solver itself has a complexity of $O(d_{rff}^3)$.
>
> > ### **Q3) Cost of Erasure**
> >
>
> We agree that the "cost of concept erasure" is a rich, multi-faceted topic. Our work focuses on what we consider to be the most fundamental axis of this cost: the direct trade-off between erasure effectiveness and task utility.
>
> Our primary goal was to first define the optimization objective of true nonlinear guardedness (full erasure) from the perspective of the trade-off, then devise an algorithm to optimize the objective, and then to rigorously **measure the degree of erasure and information leakage**. This, in turn, allows us to analyze the resulting utility-erasure trade-off, a relationship that was previously unexplored. By establishing the best known empirical trade-off, we believe our work improves our understanding of the costs of concept erasure.
>
> We agree there are **other  costs of interest (e.g., loss of perplexity).** A reliable method for utility-preserving erasure, like *Obliviator*, is a necessary prerequisite before other costs can be properly investigated.
>
> > ### **Q4) Capturing Nonlinear Statistical Dependencies and Its Advantage**
> >
>
> The goal of concept erasure is to ensure that a downstream model cannot utilize information about a sensitive attribute $S$ from a representation $X$, whether for fairness, privacy, or other reasons. Linear methods are often insufficient for this goal. For example, after applying a linear erasure method like INLP to the **BiasInBios** dataset, a simple nonlinear adversary (an MLP) can still predict $S$ from the modified $X$ with over 99% accuracy. This demonstrates that significant predictive information remains. While tractable, linear approaches can provide valuable insights, but they often fall short of the ultimate goal of complete erasure.
>
> Our use of RKHS is twofold, involves both the measurement of dependence and the architecture of the erasure process itself.
>
> **1. To Robustly Measure Nonlinear Dependencies :** This is motivated by the failure of linear correlation, as seen in the example where $X=sin(Z)$ and $S=cos(Z)$ for $Z∼\mathcal{U}(−π,π)$. The functional approach, which we adopt, asks: Are there any functions, $f$ and $g$, that can transform $X$ and $S$ so that the dependency between $f(X)$ *and* $g(S)$ *becomes linear?* For a visualization of such functions, we refer to Figure 1in [R1].
>
> The theory behind HSIC guarantees that if we search for $f$ and $g$ within an expressive RKHS (i.e., one with a characteristic kernel), and no such functions can be found, then $X$ and $S$ are truly statistically independent. We leverage this powerful guarantee in our objective (Equation 6) to learn an erasure function $ε$ that is robust against this entire expressive class of functions.
>
> **2. To Guide the Optimization via RKHS Refinement :** Expecting that a concept can be removed from the complex representation space of an LLM in a single-step optimization without distorting other useful information would be a **very strong assumption**.
> Here, we introduce our second, novel utilization of RKHS. In our RKHS Refinement step (Equation 9), we find optimal RKHS-based encoders via a constrained optimization. The constraint in this equation confines the search for these encoders to the orthogonal complement of functions that increase alignment with the sensitive attribute $S$. Within this constrained space, the objective is to find encoders that enhance the alignment with useful information from previous steps, creating a better-conditioned feature space. This makes the optimization task for the main encoder in the next iteration significantly easier and more stable, leading to a more utility-preserving erasure.
>
> > ### **Q5) Minor: implementation description of Obliviator**
> >
>
> Practical implementation details are in Appendix B. We will update this section with a PyTorch code snippet of the core algorithm, and the full code will be made publicly available for reproducibility.
>
> ---
>
> **Please consider raising the scores if we addressed your initial concerns. If you have more questions, we would be happy to answer them.**
>
> # References
> [R1] Statistical Consistency of Kernel Canonical Correlation Analysis, JMLR 2007

---

> > ### Comment · Reviewer_BjYo · 2025-08-04
> >
> > Thank you for your detailed response and for clarifying the open questions. I would like to maintain my assessment for accepting this work.

---

> > > ### Author Response · Authors · 2025-08-04
> > > **Response**
> > >
> > > Thank you for taking the time to review and engaging with our rebuttal. We appreciate your valuable feedback and insightful questions.

---

### Official Review · Reviewer_X9EX · 2025-07-03

**Clarity:** 2
**Significance:** 4
**Originality:** 4
**Rating:** 5
**Confidence:** 3

**Summary:**

The paper focuses on non-linear concept erasure. The idea is to focus on RKHS to find concepts that are non-linear. The paper provides a battery of experiments. The paper tackles an important topic and one that is of personal interest to me. The experiments are well executed. Moreover, the

**Questions:**

* Can you describe and motivate Equation 1? This is what my review will mostly hinge on.
* Can you describe the relationship with kernelized R-LACE (https://arxiv.org/abs/2201.12191) and kernelized SAL better?
* The original guardedness paper (https://arxiv.org/abs/2210.10012) formulates guardedness in terms of V-information. Does that definition hold here? Can you give such a definition to make it easier to relate to previous work?

**Ethical Concerns:**

["NO or VERY MINOR ethics concerns only"]

**Final Justification:**

This is a great paper. The authors answered my concerns.

**Limitations:**

Mentioned above.

**Paper Formatting Concerns:**

None.

**Quality:**

3

**Strengths And Weaknesses:**

I saved most of my text for this box. My fundamental problem with understanding this paper is Equation 1. Everything else derives from it. First and foremost, I am not sure what the random variables X and Y. What I would need to really understand this paper is to understand which of the variables X and Y correspond to neural representations and why. I simply don't understand why Eq.  (1) is the right adversarial game. This seems crucial to me and made it hard to engage with the rest of the paper. To be clear, I am willing to believe it is, but the paper needs to be substantially improved in this part. Beyond that, I think the contribution of the paper is to come up with a kernelization of Eq. (1).

I also wish the paper had more background on Gretton et al. -- as I understood it, there seems to be some sort of expressivity result.

---

> ### Author Rebuttal · Authors · 2025-07-31
>
> # Summary of the Review & Response
>
> We appreciate the reviewer's feedback and acknowledgement that our paper tackles an *important* topic with a *battery of experiments* that are *well executed (summary).*
>
> The reviewer's primary concerns are:
>
> 1. Motivation behind Eq. (1)(Q1)
> 2. The relationship between previously proposed methods, Kernelized SAL and R-LACE (Q2)
> 3. The connection between our framework and the V-information formalism for guardedness (Q3)
> 4. The reviewer also perceived our contribution as a "kernelization of Eq. (1)".
>
> Our responses are summarized as:
>
> 1. We clarify the motivation behind Eq. (1) through a pedagogical toy example to build intuition, showing how Eq. (1) formalizes the objective of a linear test.
> 2. We provide a detailed technical comparison for Kernelized R-LACE and kSAL, highlighting a key technical flaw in these prior methods: they rely on an unverifiable Gaussian assumption in the feature space, a limitation that our adversarial model overcomes.
> 3. We connect our HSIC-based approach to the V-Guardedness formalism, demonstrating that our guarantee of independence is stronger and more general.
> 4. We clarify that our core contribution is not merely a "kernelization”, but the design of an optimization algorithm that progressively erases the undesired concept information, with each iteration involving two steps. We explain how this solution, with its utility-preserving loss and RKHS refinement step, operationalizes the principles of HSIC for effective and consistent nonlinear erasure.
>
> # Response to Questions:
>
> > ### **Q1) Motivation Behind Eq. (1)**
> >
>
> First let us clarify the crucial point that Eq. (1) describes a linear statistical test (maximize correlation) while Eq. (6) defines an adversarial game. An adversary, using functions $f$ and $g$ from an expressive RKHS, seeks to maximize the correlation between $g(Y)$ and $f(ε(X))$. The guarding function $ε$ is simultaneously trained to minimize this correlation.
>
> To motivate the objective behind Eq. (1), we begin by the standard assumption in statistical learning:  we model $X$(learned neural representation) and $Y$ (class label like profession/sentiment) as random variables, and our data as samples from their joint distribution $P(X,Y)$. To build intuition for why Eq. (1) is a natural starting point, consider a toy example where a process generates each data pair from three independent latent variables, $Z_1,Z_2,Z_3$, each drawn from a Gaussian $\mathcal{N}(0,1)$. The observable random variables X and S are formed as follows:
>
> $$
> X=Z_1v_1+Z_2v_2\qquad Y=Z_1u_1+Z_3u_2
> $$
>
> where $v_i$ and $u_i$ are pairs of linearly independent unit vectors. In this case, any observed correlation between samples of $X$ and $Y$ is due *only* to the latent factor $Z_1$. For a dependency test we can begin by the following question :
>
> Can we find "viewpoints" that isolate the shared $Z_1$?
>
> Eq. (1) formalizes this  for the optimal linear projections $(u, v)$ that maximize correlation. For this example, the solution would be $v_1$ and $u_1$.
>
> Following this, we can also ask the inverse: Can we find viewpoints of $X$ that are independent of $Y$ ?
> Yes, if we project $X$ to orthogonal complement of $v_1$.  Therefore,  independence  requires filtering out this information (erasure) from $X$.
>
> This example illustrates how, for Gaussian RVs, the linear test in Eq. (1) is sufficient to describe their dependence structure completely.
>
> > ### **Q2) Kernelized R-LACE/SAL**
> >
>
> To clarify the relationship between these methods, we first examine how they address **linear dependencies**.
>
> The key distinction lies in the approach, not the outcome:
>
> - **R-LACE** adopts an adversarial perspective, learning a projection to thwart a linear classifier attempting to predict the sensitive attribute.
> - **SAL**, akin to the Gaussian toy example, takes a statistical approach, finding projections that nullify the linear correlation between the representation and the sensitive attribute.
>
> These are equivalent in the linear setting: zero linear correlation implies that no linear predictor can succeed. However, this only guarantees independence under approximately Gaussian distributions. For other distributions, zero correlation does not imply independence.
>
> A natural extension is to kernelize the method:
>
> 1. Map inputs X to a high-dimensional space via a Mercer feature map $\phi(X)$.
> 2. Apply a **linear** erasure method in this space.
>
> But this only protects against *linear* adversaries in that specific kernel space. As shown in Kernelized R-LACE, such protection does not generalize to other nonlinear adversaries. Like the Gaussian case, independence is only guaranteed if $\phi(X)$ is approximately Gaussian. The same limitation applies to KSAL. Statistically, optimizing against a kernel regressor only ensures *mean independence*, not full independence [R2, R3, R4].
>
> Our method addresses these limitations by using a stronger adversary. Rather than checking for linear dependence in a fixed space $\phi(X)$, we ask whether *any* nonlinear transformation $\psi(\cdot)$ exists such that a linear dependence can be found between $\psi(\phi(X))$ and $\ell(Y)$ (see Eq. 6). This effectively constructs a composite feature map $\psi(\phi(X))$, and from a statistical standpoint, optimizes for true *independence*.
>
> This nested optimization problem lacks a closed-form solution and is substantially more challenging than kernelization. We propose a more **structured and consistent optimization** framework to solve it effectively.
>
> > ### **Q3) Guardedness in terms of V-information**
> >
>
> The relationship is best understood by first considering the different, but equivalent, mathematical languages used to define **statistical independence**.  There are several equivalent ways to define statistical independence between two random variables, $X$ and $Z$:
>
> - **Probabilistic:** The conditional and marginal probabilities are the same: $P(Z|X)=P(Z)$.
> - **Information-Theoretic:** Knowing $X$ does not reduce uncertainty about $Z$. Formally, the mutual information is zero: $I(X;Z)=0$.
> - **Functional Analysis:** This view states that if no functions $f$ and $g$ can be found to observe a linear correlation between $f(X)$ and $g(Z)$ then $X$ and $Z$ are independent. (HSIC$(X,Z)=0$  with characteristic kernel)
>
> V-guardedness, is a weaker form of the information-theoretic view. It asks if a classifier from a specific family V(e.g. log-linear models) can find any predictive signal.
>
> In our approach by enforcing that HSIC $\to$ 0, we ensure a stronger condition and guard against entire space of a characteristic RKHS, not just a specific family of classifiers. To connect this directly to the language of the V-guardedness we can say:
>
> Definition (Nonlinear-Guardedness): Let $h$ be a guarding function, and let $\mathcal{F}$ and $\mathcal{G}$ be RKHSs with characteristic kernels. A representation $X$ is nonlinearly $\epsilon$-guarded by $h$ with respect to an attribute $Z$ if HSIC$(\mathcal{F},\mathcal{G},P(h(X),Z))≤ϵ$.
>
> ---
>
> # Response to Strengths and Weaknesses:
>
> > ### **The contribution of the Paper is Kernelizing Eq. (1)**
> >
>
> We appreciate the opportunity to clarify our contribution. While Eq. (1) is the linear starting point, our core novelty is the formulation of the erasure problem from a functional perspective, as defined in Eq. (9,13), and the multi-step framework we propose to solve it.
>
> First we utilize a non-linear statistical test Eq. (3) to measure full statistical dependence. To see why, consider a simple example like $X=sin(Z)$ and $Y=cos(Z)$ where $Z \sim \mathcal{U}(-\pi,\pi)$. Here, $X$and $Y$ are dependent, yet their linear correlation (Equation 1) is zero. But are there exist functions, *f* and *g*, that can transform $X$ and $Y$ so that the dependency between $f(X)$ and $g(Y)$ becomes linear? For a clear visualization of such functions for this problem, we refer to Fig. 1 in [R1]. This question is precisely what HSIC addresses. By constraining these functions $f$ and $g$ to exist in RKHS, the search for them becomes tractable. If RKHS  has a characteristic kernel, it can approximate any function arbitrarily well, then HSIC being zero implies true statistical independence.
>
> Our contribution is to adopt a functional perspective to the adversarial game, and leverage RKHS in  Obliviator, specifically in defining the objective for our RKHS refinement step Eq. (9) and the adversarial loss Eq. (13). While we use the empirical HSIC as a tool in our approach, we architect the erasure process to be more utility-preserving by:
>
> 1.  Our adversarial loss (Eq. 13) is designed to utilize HSIC for both  gradual and utility-preserving erasure.
> 2. We solve a constrained optimization in RKHS, to find optimal functions that align the feature space for the *next* iteration of erasure. This is the key to make the optimization consistent and we have shown this in our ablation study (see Fig. 9 in appendix).
>
> Therefore, our contribution is not simply kernelization of Eq. (1), our contribution is formulating and solving the erasure problem from a functional perspective.
>
> > ### **More details on Gretton et. al. and expressivity**
> >
>
> For expressivity of RKHS with a characteristic kernel see [R5]. For more mathematical details regarding the cross-covariance operator, HSIC see Appendix A.
>
> **Please consider raising the scores if we addressed your initial concerns. If you have more questions, we would be happy to answer them.**
>
> # References
>
> [R1] Statistical Consistency of Kernel Canonical Correlation Analysis, JMLR 2007
>
> [R2] Learning unbiased representations via Rényi minimization, arXiv:2009
>
> [R3] Representation learning with statistical independence to mitigate bias, WACV 2021
>
> [R4] On Characterizing the Trade-off in Invariant Representation Learning, TMLR 2022
>
> [R5]  Hilbert Space Embeddings and Metrics on Probability Measures, JMLR 2010

---

> > ### Author Response · Authors · 2025-08-04
> > **Following up on your Thoughtful Review**
> >
> > Thank you again for your time and for the thoughtful review. We have posted our full rebuttal, but wanted to provide a concise summary of our responses to your key questions:
> >
> > - **On the Motivation for Equation (1):** To build intuition for the linear case, our rebuttal uses a toy example with *Gaussian latent variables*. We explain how, for such data, the linear dependence test in Eq. (1) is sufficient to completely capture the dependency structure, establishing it as a natural and intuitive starting point for the erasure problem.
> > ----
> > - **On Comparison to Kernelized R-LACE/SAL:** Our rebuttal details the specific flaw in these prior methods: they only enforce *linear* independence within the kernel's feature space, which is insufficient for true nonlinear guardedness. Our method overcomes this by enforcing statistical independence against *nonlinear functions* within an expressive, characteristic RKHS, ensuring true nonlinear guardedness.
> > ----
> > - **On Guardedness vs. V-Information:** Our rebuttal explains that V-guardedness is a weaker condition, as it only guards against a specific, pre-defined family of classifiers. Our approach of enforcing HSIC = 0 is stronger and more general because it ensures guardedness against the entire, expressive space of functions in a characteristic RKHS.
> > ----
> > - **On the Nature of Our Contribution:**  We also explained why our method is not a simple kernelization. Our core contribution is reformulating the erasure problem from a *functional perspective*. This is necessary because linear tests fail in many cases, which we illustrate with a *sin/cos* example where *linear correlation is zero* despite clear *dependence*. Our method *operationalizes* this functional perspective using a two-step process: it uses an HSIC-based adversarial loss (Eq. 13) designed for gradual and utility-preserving erasure, combined with an RKHS refinement step (Eq. 9) that solves a constrained optimization for finding optimal encoders to align the feature space for the subsequent iteration, ensuring a stable optimization. (See Fig. 9 in Appendix for an illustration of how this step can be helpful)
> >
> > We hope these clarifications and our detailed rebuttal help address your initial concerns and encourage further engagement with our manuscript. We are available and eager to answer any other questions you might have.

---

> > ### Comment · Reviewer_X9EX · 2025-08-09
> > **thanks a lot**
> >
> > Thanks a lot! I raised my score.

---

> ### Author Response · Authors · 2025-08-07
> **Following up on Deadline Extension**
>
> Thank you again for engaging with our work and for your thoughtful review. As the discussion deadline has been extended till tomorrow, we wanted to send a brief note to say we are happy to clarify anything in our rebuttal or work before the discussion closes. We appreciate your time and consideration.

---

### Official Review · Reviewer_4SEV · 2025-07-07

**Clarity:** 3
**Significance:** 3
**Originality:** 3
**Rating:** 4
**Confidence:** 3

**Summary:**

This paper introduces Obliviator, a multi-step framework that gradually transforms the feature space to facilitate smooth and effective concept removal. It achieves this by capturing nonlinear statistical dependencies between learned representations and undesired attributes. Each iteration involves:
- Adversarial training in Reproducing Kernel Hilbert Space (RKHS): An encoder is trained to reduce dependence on the unwanted attribute while increasing dependence on the target attribute, utilizing the Hilbert-Schmidt Independence Criterion (HSIC) to enforce independence. RKHS enables capturing nonlinear dependencies in a closed form, providing a structured framework for adversarial optimization.
- RKHS Refinement: The representations are refined using RKHS functions derived from an eigenvalue problem, enhancing alignment and smoothness for the target attribute without increasing these properties for the unwanted concept. This refinement helps condition the feature space for the next iteration's adversarial training.

The paper claims several key contributions, supported by its empirical evaluation:
- Robustness to Nonlinear Adversaries
- Quantification of Erasure Cost and Empirical Upper Bound.
- Generalizability Across Language Models.

**Questions:**

- The paper acknowledges that the rigorous experimental setup is "computationally demanding". Could the authors provide a more detailed breakdown of the computational cost (e.g., average training time per iteration or per full erasure process, GPU memory footprint) for typical configurations (e.g., Frozen+Unsupervised vs. Finetuned+Supervised) and across different models/datasets used? If possible, a comparison to the computational cost of leading baselines would be highly valuable.

**Ethical Concerns:**

["NO or VERY MINOR ethics concerns only"]

**Final Justification:**

The authors have provided a comprehensive and convincing response to the feedback. I would keep the final score to boardline accept.

**Limitations:**

- The rigorous experimental setup, while contributing to quality, is acknowledged as "computationally demanding". This could limit its scalability or adoption for researchers with fewer computational resources.
- Obliviator requires "precise definitions of unwanted attributes". In real-world scenarios, such definitions may be ambiguous, unavailable, or vary culturally and ethically, potentially limiting practical applicability.
- The paper notes that pseudo-labeling, if used, "may be noisy and impact model performance, a factor not investigated". Similarly, supervised erasure "assumes access to target task labels," which might not always be available.
- Despite its advancements, the method is "still unable to quantify the minimum intervention needed for full concept erasure or determine whether erasure can occur without compromising unrelated information". This indicates a theoretical boundary that remains unaddressed.

**Quality:**

3

**Strengths And Weaknesses:**

+ Obliviator explicitly enforces statistical independence between the transformed representations and sensitive attributes by setting HSIC to zero (HSIC = 0 via Equation (6)). This is a sound theoretical approach to ensuring nonlinear guardedness, which is empirically validated to be effective, unlike prior nonlinear methods like kSAL that failed to provide true nonlinear guardedness.

+ The application of Reproducing Kernel Hilbert Spaces (RKHS) and the Hilbert-Schmidt Independence Criterion (HSIC), which are foundational to the proposed method, is explicated in detail, including the mechanism by which nonlinear statistical dependence is quantified in a closed-form solution.

---

> ### Author Rebuttal · Authors · 2025-07-31
>
> # Summary of the Review & Response
>
> We thank the reviewer for their valuable and positive feedback.  The reviewer acknowledged our *theoretical approach* to formulate the concept erasure problem and that it is *empirically validated to be effective* compared to previous nonlinear approaches (S1). The reviewer identifies that the application of *RKHS* and HSIC for measuring true statistical independence is *explicated in detail* (S2).
>
> The reviewer raised three key concerns:
>
> 1. Computational complexity and scalability (Q1, L1)
> 2. Label availability for erasure (L2, L3)
> 3. Lack of a theoretical bound on the optimal erasure trade-off (L4)
>
> These concerns align with the challenges we already discussed in the paper’s Limitations (Section 6). Our responses:
>
> 1. We clarify the central misunderstanding: the "computational demand" refers to our offline **experimental** **evaluation process**, not the *Obliviator* algorithm itself. To provide concrete evidence of *Obliviator's* practicality and scalability, we have added **tables TR1-TR5** with detailed time and memory analysis.
> 2. While this remains a broader challenge in the field, we note that recent SoTA methods increasingly rely on **pseudo-labeling** to overcome limited access to manually annotated sensitive attributes.
> 3. We clarify that while our solution is not fully closed-form, it affords stable optimization by leveraging tractable components like HSIC for dependence measure and RKHS refinement for feature space alignment.  Our experiments show *Obliviator* achieves full erasure without sacrificing utility, setting a strong benchmark for the erasure-utility trade-off.
>
> # Response to Questions:
>
> > ### **Q1) Computationally Demanding and Time/Memory Complexity BreakDown**
> >
>
> Our reference to "computationally demanding" (in the NeurIPS Checklist) refers to Obliviator’s evaluation across different datasets, models, and downstream non-linear classifiers. We acknowledge that this is uncommon and may have caused confusion. As we demonstrate below, Obliviator itself is computationally very efficient.
>
> As detailed in Appendix C (“Probing Networks”), our evaluation is intentionally rigorous. To robustly measure information leakage, we train 10 distinct probing classifiers, among which the training of hard-margin SVMs was the most computationally intensive component. We report the maximum accuracy across all probes as a worst-case leakage estimate.
>
> This exhaustive evaluation is performed offline to validate our method's effectiveness, and its cost is entirely separate from the operational cost of the erasure algorithm. We believe such thorough validation is crucial for making reliable claims about the concept erasure problem.
>
> Tables [TR1] and [TR2] below detail the time and memory complexity of our method. All benchmarks were run on a single NVIDIA RTX-A6000 GPU, using the full BiasInBios dataset (250K samples) in a single batch.
>
> ### **TR1. Time Complexity**
>
> | Time  Complexity | Ads Training (Per Epoch) | RKHS Ref |
> | --- | --- | --- |
> | Theoretical | $O(Nd_{embd}d_{hidden} + Nd_{rff}d_{hidden}+ Nd^x_{rff}d^s_{rff}+Nd^x_{rff}d^y_{rff})$ | $O(d_{rff}^3 + Nd^x_{rff}d^s_{rff}+Nd^x_{rff}d^y_{rff})$ |
> | BERT | 177.2 ms ± 0.38 ms | 172 ms ± 1.23 ms |
> | DeepSeek | 219.6 ms ±1.4680 ms | 165 ms  ±  1.65 ms |
> ||||
>
> ### **TR2. Memory Complexity**
>
> | Memory  Complexity | Ads Training | RKSH Ref |
> | --- | --- | --- |
> | Theoretical | $O(d_{embd}d_{hidden} + d_{rff}d_{hidden} +Nd_{rff}+d_{rff}^2)$ | $O(d_{rff}^2 + Nd_{rff} )$ |
> | Utilized Memory | 8.18GB | 6.78GB |
> |  |  |  |
>
>
> The table below reports the time required to achieve specific gender accuracy targets, with results broken down by language model, dataset, and erasure setting. (Random Chance=RC)
>
>
> ### **TR3-Bias in Bios Frozen**
>
> | ***BERT*** | 99.4%→70% (sec) | 70%→65%(sec) | 65%→60% (sec) | 60%→RC (sec) | 99.4%→RC (sec) |
> | --- | --- | --- | --- | --- | --- |
> | Unsupervised | 21.9±0.05 | 27.4±0.06 | 107.76±0.26 | 323.7±0.7 | 482.9± 1.1 |
> | Supervised | 27.1±0.06 | 72.26±0.1 | 153.5±0.3 | 487.7±0.5 | 740.7± 1.7 |
> | ***DeepSeek*** |  |  |  |  |  |
> | Unsupervised | 74.3±0.5 | 67.53±0.46 | 94.54±0.64 | 189.08$\pm$1.28 | 425.4±2.9 |
> | Supervised | 156 ±1 | 100.3±0.67 | 390.1±2.63 | 1070$\pm$7.2 | 1716.3 ±11.6 |
> |||||||
>
> ### **TR4-Dial Mention Frozen**
>
> | ***BERT*** | 80%→70% (sec) | 70%→60% (sec) | 60%→RC (sec) | 80%→RC (sec) |
> | --- | --- | --- | --- | --- |
> | Unsupervised | 55.2±0.1 | 193.3±﻿0.5 | 1293.2 ±﻿3.0 | 1541.67±﻿3.6 |
> | Supervised | 126.4±﻿0.3 | 144.5±﻿0.3 | 1372.9±﻿3.7 | 1643.8±3.8 |
> ||||||
>
> ### **TR5-Bias in Bios Finetuned**
>
> | BERT | 99.4%→70%(sec) | 70%→65% (sec) | 65%→63% (sec) | 63%→RC (sec) | 99.4%→RC (sec) |
> | --- | --- | --- | --- | --- | --- |
> | Unsupervised | 37.7±﻿0.08 | 62.8±﻿0.14 | 540.7±1.2 | 4225.6±9.35 | 4866.7±9.6 |
> | Supervised | 43.2±﻿0.1 | 162.06±﻿0.36 | 108.04±0.24 | 5596±12.4 | 5909±12.2 |
> |||||||
> ---
> ## **Comparison with Baselines**
>
> ### **TR6- FARM Bias in Bios gender(acc)=62.91**
>
> | Batch Size | Memory(MB) | Time( sec per epoch) | Total Time(sec) |
> | --- | --- | --- | --- |
> | 1024 | 96.4 | 21.4±1.2 | 535±30 |
> | 4096 | 305.44 | 13.22±1.03 | 330.5±25.75 |
> |||||
>
> ### **TR7-KRAM Bias in Bios gender(acc)=68.45**
>
> | Batch Size | Memory(MB) | Time(sec per epoch) | Total Time(sec) |
> | --- | --- | --- | --- |
> | 1024 | 76.8 | 65.58±1.4 | 3279±65.2 |
> | 4096 | 479.04 | 45.81±8.5 | 2290±42.5 |
> |||||
>
> ---
> # Response to Limitations:
>
> > ### **L1) Adaptation for research**
> >
>
> We wish to respectfully clarify this point further. The "computationally demanding" aspect refers exclusively to the **offline evaluation protocol** we used to validate our results, not our proposed algorithm, Obliviator.
>
> To make this distinction unambiguous, **Tables TR1-TR2** provides a detailed breakdown of the elapsed time for each step. As the table demonstrates, the runtime of Obliviator is highly practical. The intensive evaluation suite, while crucial for making reliable scientific claims, is not a required step for applying our method. The computational requirements of Obliviator itself do not pose a barrier to its adoption for research and scalability.
>
> > ### **L2) Precise Definition of Unwanted Attributes**
> >
>
> We agree and discuss this very point in our **Limitations (Section 6, Page 9)**, acknowledging that defining unwanted attributes is challenging and context-dependent.
>
> Any concept erasure method fundamentally requires a precise definition of the concept to be removed. In the context of our paper, all discussed baselines rely on sensitive attribute labels for erasure. The challenge of defining these concepts is analogous to the often subjective task of defining fairness criteria in ML.
>
> Our work focuses on the crucial next step: *given* a defined concept, how to erase it with maximum efficacy. While we note future work could aid the definition phase (e.g., via pseudo-labeling ), our contribution is in advancing the erasure mechanism itself.
>
> > ### **L3)  Label Availability**
> >
>
> We agree that the availability and quality of labels, for both the unwanted concept and the target task, are crucial challenges, which we acknowledge in our Limitations section (Section 6).
>
> We believe that existing pseudo-labeling techniques, facilitated by more powerful foundation models, offer a viable path to address these challenges. Recent literature provides strong evidence for this. For instance, [R1] use an **optimal transport framework** to handle uncertainty in pseudo-labels as part of an augmented training phase, improving performance in low-resource settings. Similarly, [R2] use pseudo-labels for debiasing and show in their Table 2 that this approach is highly competitive with models trained on true labels.
>
> These methods demonstrate that robust solutions for label generation under uncertainty are actively being developed. In contrast, our work focuses on seeking the optimal/near-optimal erasure mechanism that can operate effectively on labeled data, whether they are ground-truth or high-quality pseudo-labels.
>
> > ### **L4) Minimum Intervention**
> >
>
> We agree and explicitly state in our **Limitations (Section 6)** that our method is "still unable to quantify the minimum intervention needed for full concept erasure". To our knowledge, a closed-form, theoretical guarantee for optimal nonlinear erasure remains an open problem in the field.
>
> While a theoretical optimum is yet to be found, our work focuses on studying the utility-erasure trade-off, and establishing a strong **empirical upper bound** on this trade-off. For instance, **Figs. 3a, 3b, and 6b** show scenarios where our method achieves nearly full erasure of the unwanted attribute while preserving a high degree of task accuracy, indicating its practical ability to preserve unrelated information during erasure.
>
> The effectiveness of our approach stems from its design. While the main optimization (Eq. 6) does not have a closed-form solution, we make the optimization more tractable by:
> 1.  Using  HSIC to capture nonlinear dependencies in a closed form.
>
> 2.  An iterative process, each with two-steps that refines the feature space via an RKHS-based EVP (Eq. 11), which facilitates smoother erasure and more consistent optimization.
>
> The significant impact of the two-step process is demonstrated in our ablation study in **Appendix D.1 (Fig. 9)**, where we observe erasure by a single step with encoder is less stable and utility-preserving, unlike *Obliviator*. This provides strong evidence that our solution, while not offering a theoretical guarantee, represents a significant step towards achieving optimal erasure in practice.
>
> ---
>
> **Please consider raising the scores if we addressed your initial concerns. If you have more questions, we would be happy to answer them.**
>
> # References
>
> [R1] FFB: A Fair Fairness Benchmark for in-processing group fairness methods,  ICLR2024
>
> [R2] FairerCLIP: Debiasing CLIP's Zero-Shot Predictions using Functions in RKHSs, ICLR2024

---

> > ### Author Response · Authors · 2025-08-04
> > **Following up on your Thoughtful Review**
> >
> > Thank you again for your time and for the thoughtful and detailed review. We have posted our full rebuttal, but wanted to provide a concise summary of our responses to your key concerns:
> >
> > - **On Computational Cost:** We'd like to clarify a key point. The "computationally demanding" aspect refers exclusively to our rigorous *evaluation protocol*, not the *obliviator* algorithm itself. To demonstrate the algorithm's efficiency, we provide detailed benchmarks showing that, for example, on a single GPU each encoder *training epoch takes ~177ms* and the subsequent *RKHS refinement (EVP) step takes ~172ms*.
> > ----
> > - **On Practical Limitations (Label Definition & Availability):** We agree that defining sensitive attributes and ensuring label availability are crucial, real-world challenges. We position our work as providing the core erasure mechanism once an attribute is defined. We discuss two recent works demonstrating that *pseudo-labeling techniques* offer a viable path to limited label availability for post-hoc erasure.
> > ----
> > - **On Theoretical vs. Empirical Bounds on Erasure:** We agree that a closed-form, theoretical guarantee for optimal nonlinear erasure remains an open problem. Our work's key contribution is to establish a strong benchmark on this utility-erasure trade-off. Our method's design, using *HSIC* and the *RKHS refinement step*, creates a stable optimization. This design allows our method, in certain scenarios, to achieve nearly full erasure while preserving a high degree of task accuracy, as shown in the results for specific datasets and models. (see  Figs. 3a, 3b, and 6b)
> >
> > We hope this summary and our full rebuttal have successfully addressed your concerns. We are available and eager to answer any further questions you might have before the discussion period ends.

---

> ### Comment · Reviewer_4SEV · 2025-08-05
> **Response**
>
> The authors have provided a comprehensive and convincing response to the feedback. I would keep the final score as boardline accept.

---

> > ### Author Response · Authors · 2025-08-06
> > **Response**
> >
> > Thank you for engaging with our rebuttal and for your valuable feedback. We are glad that we have addressed all of your questions and are happy to answer any final questions you may have before the deadline

---

### Note · Authors · 2025-08-13

We thank the reviewers for their insightful feedback and the program committee for a helpful review process. We are confident that our rebuttals and the resulting discussions successfully addressed all reviewers' concerns.

The discussion period concluded with the following positive final comments from reviewers:

- **Reviewer eWgU:** "I appreciate the response that fully answers my concerns. I hope the new results -- and particularly the fairness metrics -- will be incorporated in the final version. I increased my score accordingly."
- **Reviewer BjYo:** "Thank you for your detailed response and for clarifying the open questions. I would like to maintain my assessment for accepting this work."
- **Reviewer 4SEV:** "The authors have provided a comprehensive and convincing response to the feedback. I would keep the final score as borderline accept."
- **Reviewer X9EX:** "Thanks a lot! I raised my score."

We are grateful for the reviewers' engagement, which allowed us to strengthen our work. Our work, to the best of our knowledge, is the first to study the utility-erasure trade-off in Concept Erasure, a vital field for making language model more trustworthy. Moreover, our method achieves the best-known trade-offs in this area. Through the discussion period, we:
-   Clarified our contribution: an iterative solution that employs a functional perspective on erasure. Each iteration involves two steps: (1) an adversarial optimization with a utility-preserving loss (Eq. 13), and (2) a constrained optimization in RKHS (Eq. 9) that aligns the feature space to stabilize the next adversarial step.
- Provided new results showing improved downstream fairness and performed erasure on an imbalanced dataset to explore its effect on trade-off.
- Performed ablation studies to validate the effectiveness of elements involved in our two-step solution and to analyze its complexity, confirming its practicality.
- Clarified that the observed trade-off is a diagnostic of the model's learned dependencies and not necessarily a true causal link. We pointed to the milder trade-off observed in newer models like DeepSeek compared to BERT (Fig. 6).
- Detailed the flaws in prior kernelized methods and showed how our functional approach generalizes to guard against nonlinear adversaries in an expressive RKHS, not just a specific subset as in prior work.

We believe the review process has validated our work and its contribution to the field. Thank you for your time and consideration.

---

### Decision · Program_Chairs · 2025-09-17

**Decision:**

Accept (poster)

**Comment:**

This paper addresses the problem of concept erasure, i.e., removing sensitive or unwanted attributes from learned representations, while remaining robust to nonlinear adversaries. The proposed method, Obliviator, is a post-hoc, multi-step framework combining HSIC-based adversarial optimization in RKHS with a refinement step to recondition the feature space for stability and utility preservation. This approach captures nonlinear dependencies in closed form, enforces statistical independence against a broad class of nonlinear adversaries, and quantifies the utility-erasure trade-off, establishing an empirical upper bound on this relationship.

Reviewers noted that the problem is important, the formulation is well-motivated, and the evaluation is thorough, spanning multiple datasets, erasure scenarios (supervised/unsupervised; frozen/fine-tuned), and representations from both older and more recent language models. The method showed consistent gains over baselines, robustness to strong adversaries, and better utility preservation at equivalent levels of erasure. Reviewer questions about theoretical grounding, relation to prior work, fairness implications, ablations, and computational cost were addressed through additional explanations, experiments, and benchmarking. Overall, the work is viewed as making both a conceptual contribution by recasting erasure from a functional RKHS perspective and providing a strong empirical benchmark for nonlinear guardedness.